# Differential modulation of haematopoietic and oxidative injury by PARP-1 and ATR kinase inhibition in a murine model of acute irradiation

**Baydaa Taher Sih**⊙*

Department of Physics, College of Science, University of Baghdad, Baghdad, Iraq

* baydaa.sih@sc.uobaghdad.edu.iq

## Abstract

Temporal Modulation of Acute Radiation Injury by Post-Exposure PARP-1 versus ATR Kinase Inhibition.

**Editor:** Hesham M.H. Zakaly, Ural Federal University named after the first President of Russia B N Yeltsin Institute of Physics and Technology: Ural'skij federal'nyj universitet imeni pervogo Prezidenta Rossii B N El'cina Fiziko-tehnologiceskij institut, RUSSIAN FEDERATION

## Background

PARP-1 and ATR inhibitors have been employed as radiosensitizers in cancer therapy. In addition to this, there is little evidence about the role of these chemicals after a dose of radiation, specifically in relation to reducing damage to healthy cells rather than increasing it.

## Objective

The goal of this study was to determine if PARP-1 or ATR kinase inhibitors, when delayed administered, could decrease the severity of the condition caused by an acute dose of radiation and compare the impact of PARP-1 and ATR inhibitors.

## Methods

Mice of the C57BL/6 strain were subjected to a total body irradiation dose of 2.5 Gy γ-rays. In each experiment, thirty minutes following exposure to IR, either the PARP inhibitor, olaparib at 50 mg/kg or the ATR kinase inhibitor VE-821 at 25 mg/kg was injected intraperitoneally into the mice. The parameters of blood were measured at the same time as oxidative stress, including the activity of superoxide dismutase and levels of malondialdehyde. Examination of DNA damage and repair dynamics was done through the use of γ-H2AX immunofluorescence.

## Results

Following exposure to ionising radiation, mice which had received a PARP inhibitor showed less marked haemoglobin reduction and less radiation-induced leucopenia than control mice exposed to radiation alone. The modifications were accompanied

**Data availability statement:** All relevant data are within the manuscript and its Supporting Information files.

**Funding:** The author(s) received no specific funding for this work.

**Competing interests:** The authors have declared that no competing interests exist.

by a decrease in lipid peroxidation plus enhanced antioxidant activity. This observation that PARP inhibition enhances repair of DNA double-strand breaks is consistent with earlier data from other researchers. In contrast the protective effects of the ATR kinase inhibitor were not apparent when administered after the DNA damage had been inflicted in any of the parameters examined.

## Conclusion

While administering PARP inhibitors following the exposure to ionising radiation was found to reduce the severity of the condition experienced by the subject, as a result of acute radiation syndrome, the same was not seen when the treatment with ATR inhibitors was given to affected subjects. The results of this study show the significance of administering therapeutic agents at the right time to stop the injury to normal cells when DNA repair pathways are activated by exposure to ionising radiation.

## 1. Introduction

Therapeutic and accidental exposure to ionizing radiation triggers a complex, time-evolving injury that extends far beyond the initial DNA lesions. Radiation therapy stands as a fundamental treatment in clinical oncology but its effectiveness becomes restricted because of the toxic effects which occur when high doses of radiation damage quickly dividing normal cells including bone marrow cells during total or partial body irradiation [1]. The situation requires immediate work on treatment protocols which need to protect healthy tissues from damage while preserving their cancer-fighting abilities.

The DNA double-strand break (DSB) represents the most severe radiation-caused DNA damage which the DNA damage response (DDR) system controls through its complex protein network [2]. The DDR contains its first measurable event which occurs when histone H2AX undergoes serine-139 phosphorylation to form γ-H2AX foci which become visible under a microscope at DSB locations [3]. The quick appearance of γ-H2AX serves as a reliable indicator for DSB formation but scientists have not determined what causes its prolonged existence and disappearance in cells. The disappearance of γ-H2AX foci is conventionally associated with successful DNA repair [4]. Research shows that γ-H2AX signaling which continues beyond the first 24 hours indicates cells have started additional responses which include senescence and ongoing oxidative damage and metabolic problems and survival of cells that remain relatively intact [5,6]. Scientists need to assess γ-H2AX status as a biomarker because its complex characteristics exist when DDR modulation therapies are employed.

The therapeutic application of DDR kinase inhibitors which includes PARP and ATR kinase has transformed cancer therapy because these drugs make cancer cells more sensitive to radiation [7,8]. The DNA repair process of single-strand break repair becomes impaired when PARP enzymes get trapped by Olaparib and other PARP inhibitors which results in BRCA-deficient tumor cell death [9]. The replication

stress response master regulator ATR inhibitor (ATRis) blocks the cell cycle progression which results in mitotic catastrophe for cells that maintain their DNA damage [10]. Research studies confirm the canonical paradigm which demonstrates that radiation therapy produces better results when patients receive these inhibitors because the inhibitors lengthen the period when early γ-H2AX foci appear (within 0–24 hours) which serves as a pharmacodynamic biomarker for radiosensitization [11,12]. This paradigm, however, is built largely on measurements taken during the acute phase of the DDR. The scientific community lacks understanding about how delaying DDR inhibitor treatment affects biological processes which start after cells detect damage and begin their repair process. The time-dependent factor enables scientists to create radiation injury treatment protocols and develop complex radiotherapy plans which safeguard various levels of healthy tissue.

Research shows that PARP inhibition makes cells more responsive to radiation therapy when used together with radiation but the timing of PARP inhibition after radiation exposure leads to different biological responses. PARP-1 activation beyond its normal range following DNA damage exposure causes NAD$^+$ substrate depletion which results in parthanatos through ATP depletion and mitochondrial breakdown [13,14]. The start of PARP inhibitor therapy for metabolic crisis serves to defend cellular energy stability. The research provides a possible experimental question which investigates how delayed PARP inhibition treatment affects the duration of γ-H2AX signals in cells.

The main function of ATR exists to protect damaged replication sites and activate S and G2/M phase checkpoints which makes its blocking action independent from the proposed metabolic pathway [15]. Scientists can determine which elements trigger delayed DDR signaling through their research into these two distinct pathways using time-based studies.

The researchers monitored DNA damage signals in mice who received total body radiation to identify the first damage indicators which appeared before tissue damage became visible. The research team expected that PARP inhibition after radiation exposure would result in quicker disappearance of late γ-H2AX foci but ATR inhibition would not produce this outcome. The research findings demonstrated that PARP inhibition after radiation exposure resulted in quicker disappearance of late γ-H2AX foci which also reduced oxidative stress and hematopoietic damage instead of accelerating DNA repair.

The study examines how DNA damage signaling mechanisms from metabolic stress responses react to ionizing radiation through PARP and ATR pathway blocking at various time points. Using multiparametric longitudinal analysis in a murine model, we will:

The research investigates how Olaparib as a PARP inhibitor and VE-821 as an ATR inhibitor delay their administration and affect γ-H2AX persistence duration following radiation exposure throughout 72 hours. The research needs to establish if late γ-H2AX resolution patterns match the changes which occur in oxidative stress indicators (MDA, SOD) and blood cell production (WBC, Hb).

The research investigates how PARP inhibition is delayed because it reduces γ-H2AX persistence, which prevents NAD$^+$-dependent metabolic breakdown and subsequent oxidative stress from occurring, rather than speeding up the initial DNA repair process.

The research will create a system which enables scientists to understand DDR biomarkers through time-based analysis while demonstrating the best treatment approaches for radiation therapy in animal studies.

## 2. Materials and methods

### 2.1. Study design and temporal rationale

The research used time-based multi-parameter testing to analyze how delayed drug treatment of essential DNA damage response pathways affects biological responses after receiving sudden ionizing radiation. The main research question predicted that post-exposure treatment methods would produce different responses based on which biological pathway they focused on: Blocking PARP 1 activity was predicted to prevent metabolic damage that occurs when NAD$^+$ levels decrease while ATR kinase blockade was predicted to disrupt cell replication stress response mechanisms. The research

analyzed injury progression and first-stage recovery patterns across three time stages, which covered 2–48 hours and 7 days and it assessed functional recovery through 30-day survival rates [1,3,13].

## 2.2. Animals, ethics, and husbandry

All animal experiments were conducted in strict accordance with international guidelines for the care and use of laboratory animals and were approved by the College of Science Research Ethics Committee, University of Baghdad (Reference Nos. CSEC/1125/01244 and CSEC/1129/0139; 24 February 2023). A total of 80 specific pathogen free C57BL/6 mice (40 males, 40 females), aged 8–10 weeks and weighing 20–25 g, were obtained from the Animal House Unit, University of Baghdad. The mice spent one week under controlled environmental conditions before the study began at 22 ± 2 °C and 55 ± 10% humidity with a 12 h light/dark cycle, while they received standard chow and filtered water without restriction. The researchers tracked animal health and behavior and psychological state at all times during the entire study period. Fig 1 illustrates representative images of the animals in the housing facility and during experimental procedures.

Sample size was calculated a priori using G*Power 3.1 [16]. The study requires at least ten animals for each group because researchers expect to find a significant effect size of f = 0.8 while maintaining a power of 1 β = 0.90 for white blood cell count and 30 day survival endpoints at α = 0.05. The study required 80 mice to establish eight experimental groups which contained ten mice in each group. A computer generated randomisation sequence, stratified by sex and body weight, was used to assign animals to groups. The team executed all procedures to minimize animal suffering while working with the smallest possible number of animals.

The mice received human euthanasia through cervical dislocation after receiving deep anesthesia from ketamine (100 mg/kg) and xylazine (10 mg/kg) injections into their peritoneal cavity at the scheduled time points. The method provides immediate permanent loss of consciousness which follows the American Veterinary Medical Association standards for rodent euthanasia.

## 2.3. Irradiation procedure

Whole body γ irradiation was delivered using a calibrated $^{137}$Cs Gamma Cell 3000 irradiator (Nordion, Canada). The PMMA restrainers of well ventilated custom built design held unanesthetized mice who received a 2.5 Gy dose at 0.02

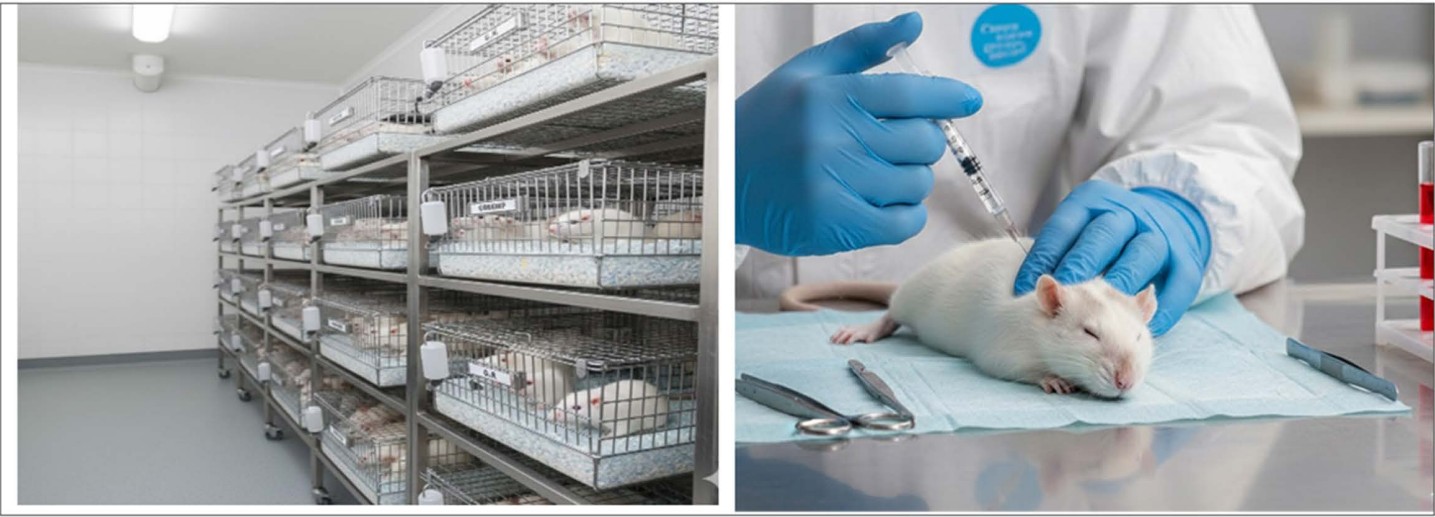

**Fig 1. Illustrates representative images of the animals in the housing facility and during experimental procedures.**

Gy/s. The irradiation chamber contained TLD 100 dosimeters which verified dosimetry through measurements that showed dose uniformity within a 5% margin [13]. Sham irradiated control animals underwent identical handling and restraint but were not exposed to the radioactive source [3].

### 2.4. Post irradiation pharmacological interventions

The researchers tested the delayed intervention hypothesis by starting all inhibitor treatments thirty minutes following radiation exposure which matched the timeframe of actual medical countermeasure deployment [16]. The PARP inhibitor Olaparib (Selleck Chemicals) required dissolution in DMSO to create a 25 mg/mL stock solution which then got diluted in sterile PBS to achieve a 2.5 mg/mL working solution. The solution received intraperitoneal (i.p.)injection at a dose of 50 mg/kg body weight [11].

The ATR inhibitor VE 821 (MedChemExpress) needed dissolution in sterile PBS to make a 1.25 mg/mL solution which was then used for i.p. administration. The study used 25 mg/kg as the dose for this treatment [17]. Vehicle control animals received either DMSO/PBS (matching the Olaparib vehicle) or PBS alone. The solutions received their preparation under sterile conditions right before the administration process. The researchers chose their dose amounts and treatment schedule based on previous radiobiology studies which demonstrated the established pharmacokinetic and efficacy profiles [17,18].

### 2.5. Experimental groups

The researchers distributed mice into eight different experimental groups which contained ten mice in each group.

1. Untreated Control – no irradiation, no vehicle/inhibitor.

2. Vehicle Control (DMSO/PBS) – no irradiation, DMSO/PBS i.p.

3. Vehicle Control (PBS) – no irradiation, PBS i.p.

4. PARP Inhibitor Only – no irradiation, Olaparib 50 mg/kg i.p.

5. ATR Inhibitor Only – no irradiation, VE 821 25 mg/kg i.p.

6. Radiation Only – 2.5 Gy + vehicle i.p.

7. Radiation + PARP Inhibitor (R+PARPi) – 2.5 Gy + Olaparib 50 mg/kg i.p.

8. Radiation + ATR Inhibitor (R+ATRi) – 2.5 Gy + VE 821 25 mg/kg i.p.

The detailed information about all experimental groups appears in Table 1, which shows treatments, doses and administration timing and animal numbers for each time point.

### 2.6. Longitudinal monitoring and survival analysis

The research team performed two daily animal inspections which spanned thirty days beginning on the day when radiation exposure occurred. The clinical scoring system used four criteria to assess morbidity which included weight loss greater than 20% of the initial body weight and hunched posture and reduced activity and piloerection. Mice meeting predefined humane endpoints were euthanized immediately, and the date was recorded as a mortality event. The Kaplan–Meier method analyzed survival data to determine survival outcomes. The log rank (Mantel–Cox) test evaluated differences between the three treatment groups which included Radiation Only and R+PARPi and R+ATRi.

**Table 1. Experimental groups, treatments, and animal allocation.**

| Group | Irradiation (2.5 Gy) | Treatment | Dose | Route | Timing (post-IR) | n (per time point) | Total n |
|---|---|---|---|---|---|---|---|
| 1. Untreated Control | No | – | – | – | – | 5 | 10 |
| 2. Vehicle (DMSO/PBS) | No | DMSO/PBS | – | i.p. | 30 min | 5 | 10 |
| 3. Vehicle (PBS) | No | PBS | – | i.p. | 30 min | 5 | 10 |
| 4. PARP Inhibitor Only | No | Olaparib | 50 mg/kg | i.p. | 30 min | 5 | 10 |
| 5. ATR Inhibitor Only | No | VE-821 | 25 mg/kg | i.p. | 30 min | 5 | 10 |
| 6. Radiation Only | Yes | Vehicle | – | i.p. | 30 min | 5 | 10 |
| 7. Radiation + PARPi | Yes | Olaparib | 50 mg/kg | i.p. | 30 min | 5 | 10 |
| 8. Radiation + ATRi | Yes | VE-821 | 25 mg/kg | i.p. | 30 min | 5 | 10 |
| **Total** | **80** | | | | | | |

IR: irradiation; i.p.: intraperitoneal; PARPi: PARP inhibitor; ATRi: ATR inhibitor.

## 2.7. Multi-timepoint biospecimen collection

The researchers performed tissue collection on five randomly chosen animals from each group at each scheduled time point. Blood and tissue sampling were performed as follows:

The research includes three blood tests which measure hematology and plasma biochemistry at 24 hours and 72 hours and 7 days following radiation exposure.

The research needs to assess oxidative stress markers at 48 hours following radiation exposure because this time point shows the peak oxidative stress levels.

The research includes DNA damage kinetics measurements which were conducted at three different time points following radiation exposure at 2 h, 24 h and 48 h.

The research design followed a longitudinal pattern which reduced animal participation while it allowed scientists to study injury and recovery processes through kinetic measurements [19].

## 2.8. Haematological analysis

Whole blood was collected from the retroorbital plexus under light isoflurane anaesthesia and analyzed within 2 h using an automated Sysmex XP 300 haematology analyzer. The researchers documented three blood parameters which consisted of total white blood cell (WBC) count and haemoglobin (Hb) concentration and platelet count [20].

## 2.9. Oxidative stress and antioxidant capacity

Liver and spleen homogenates were prepared on ice. The TBARS assay determined lipid peroxidation levels by measuring malondialdehyde (MDA) absorption at 532 nm to obtain its results [21]. Superoxide dismutase (SOD) activity was determined by inhibition of cytochrome c reduction at 550 nm; one unit of SOD was defined as the amount of enzyme causing 50% inhibition [22].

## 2.10. DNA damage assessment and target engagement validation

γ H2AX immunofluorescence: Formalin fixed, paraffin embedded spleen sections were stained with anti γ H2AX (Ser139) antibody (Cell Signaling Technology, 1:500) and counterstained with DAPI. γ H2AX foci were enumerated in >100 nuclei per sample using a Leica DMi8 fluorescence microscope. Three time points (2 h, 24 h, and 48 h post irradiation) were analysed [16,23].

Western blotting: Snap frozen spleen samples from the 24 h cohort were lysed and probed with:

- Anti poly(ADP ribose) (PAR) antibody to verify PARylation suppression in the R+PARPi group [15].

Anti-phospho-CHK1 (Ser345) antibody was used to verify ATR pathway inhibition in the R+ATRi group. These pharma-codynamic assessments provided direct evidence of on target drug activity [24,25].

### 2.11. Statistical analysis

The data presentation shows mean values with standard deviation (SD) and median values with interquartile range (IQR) for each measurement. Normality and homogeneity of variances were assessed using the Shapiro–Wilk test and Levene's test, respectively [26]. For time course data (WBC, Hb, γ H2AX), two way repeated measures analysis of variance (ANOVA) with treatment and time as factors was performed, followed by Tukey's honest significant difference (HSD) post hoc test [27]. Single time point comparisons (MDA, SOD at 48 h) were analysed using one way ANOVA with Tukey's HSD. Survival curves were compared using the log rank test [18]. Correlations between parameters were examined using Pearson's or Spearman's correlation coefficients, as appropriate. Effect sizes for significant between group differences are reported as Cohen's d. All statistical analyses were conducted using IBM SPSS Statistics version 28.0 and GraphPad Prism version 9.0 [26,27].

## 3. Results and discussion

### 3.1. White blood cell count (WBC x10³/µL)

The white blood cell numbers in Table 2 show that radiation exposure caused a major decrease, which brought the cell count down to $4.40 \pm 0.52 \times 10^3/\mu L$ from $6.50 \pm 0.61 \times 10^3/\mu L$ within 24 hours. The WBC count decreased by 32.31% (p < 0.001) from its initial value. The fast decline in cell numbers follows the known radiation-induced DNA damage process, which leads to apoptosis in hematopoietic stem and progenitor cells [28]. The results show acute leukopenia, but the study cannot show how the blood cells will recover because it only measured at one time point.

PARP Inhibition: Protective Effects with Mechanistic Uncertainties

The combination of PARP inhibition with radiation attenuated the decrease in WBC counts to 19.23%, which was significantly less severe than the 32.31% reduction observed in the radiation-only group (p = 0.008; Table 3). The protective mechanism of PARP inhibition could result from multiple biological pathways which operate in the body.

The activation of PARP results in NAD+ depletion which causes proliferating hematopoietic cells to lose their energy supply [29].

Table 2. White Blood Cell Count (WBC × 10³/µL) – 24 Hours after-Irradiation.

| Experimental Group | Mean ± SD | 95% CI | P-value & Control | P-value &Radiation Only |
|---|---|---|---|---|
| Control | 6.50 ± 0.61 | [6.12, 6.88] | — | — |
| Vehicle (DMSO) | 6.42 ± 0.58 | [6.05, 6.79] | 0.782 | — |
| Vehicle (PBS) | 6.38 ± 0.55 | [6.03, 6.73] | 0.615 | — |
| PARP Inhibitor Only | 6.45 ± 0.59 | [6.07, 6.83] | 0.540 | — |
| ATR Inhibitor Only | 6.38 ± 0.62 | [5.98, 6.78] | 0.320 | — |
| Radiation Only | 4.40 ± 0.52 | [4.08, 4.72] | <0.001 | — |
| Radiation + Vehicle | 4.52 ± 0.55 | [4.18, 4.86] | <0.001 | 0.00452 |
| Radiation + PARP Inhibitor | 5.25 ± 0.58 | [4.89, 5.61] | <0.001 | 0.008 |
| Radiation + ATR Inhibitor | 4.45 ± 0.54 | [4.12, 4.78] | <0.001 | 0.002 |

*One-way ANOVA: F(8,63)=15.8, p<0.0001*

**Table 3. Percentage Difference Analysis – White Blood Cell Count.**

| Experimental Group | % Change vs Control | % Change vs Radiation Only | Clinical Significance |
|---|---|---|---|
| **Non-Irradiated Groups** | | | |
| Control | 0% | — | Normal |
| Vehicle (DMSO) | −1.23% | — | |
| Vehicle (PBS) | −1.85% | — | |
| PARP Inhibitor Only | −0.77% | — | |
| ATR Inhibitor Only | −1.85% | — | |
| **Irradiated Groups** | | | |
| Radiation Only | −32.31% | 0% | Severe Leukopenia |
| Radiation + Vehicle | −30.46% | +2.73% | Severe Leukopenia |
| Radiation + PARP Inhibitor | −19.23% | +19.32% | Moderate Leukopenia |
| Radiation + ATR Inhibitor | −31.54% | +1.14% | Severe Leukopenia |

- PARP inhibition may shift the mode of cell death from necrosis toward apoptosis [30].

The treatment process might create a short-term dormant period in blood cell development cells which would make them less sensitive to radiation [31].

The researchers have proposed several possible mechanisms, but these remain unproven because the study failed to measure NAD$^+$ levels, PARylation activity, and apoptotic markers in bone marrow tissue.

Despite the acknowledged methodological limitations, the present findings carry important translational implications that extend beyond descriptive DDR signaling. The divergent biological outcomes observed following delayed PARP versus ATR inhibition underscore timing as a decisive variable in DDR pathway modulation, rather than target selection alone [9,10,30]. These observations suggest that therapeutic scheduling may represent an underappreciated dimension for optimizing the therapeutic ratio between tumor eradication and normal tissue preservation [1,2,13]. Conceptually, a rational treatment paradigm could involve early administration of DDR inhibitors to exploit synthetic lethality in rapidly proliferating tumor cells, followed by delayed intervention designed to attenuate prolonged metabolic and signaling stress in slowly dividing normal tissues [12,30,32]. Such a strategy, while theoretically compelling, requires rigorous validation in well-controlled in vivo tumour-bearing models that permit simultaneous assessment of tumor control efficacy and normal tissue toxicity, particularly in radiosensitive organs such as bone marrow and intestinal epithelium [13,28].

The radiation-only group exhibited exceptionally wide dispersion in γ-H2AX responses, with a standard deviation approximating 67% of the mean, exceeding what would be expected from stochastic biological variation alone. This degree of heterogeneity strongly suggests the coexistence of at least two functionally distinct cellular subpopulations: one exhibiting heightened radiosensitivity and sustained DDR signaling, and another displaying relative resistance to radiation-induced damage [3,4,8]. Such bimodal response behavior is consistent with established models of cellular heterogeneity driven by differences in cell-cycle phase distribution, replication status, and metabolic resilience at the time of irradiation [3,10]. Cells residing in S or G2 phases, or those with elevated metabolic capacity, are known to exhibit fundamentally distinct DDR kinetics compared to quiescent or metabolically compromised counterparts [10]. Formal validation of this interpretation would require single-cell–resolved analyses, which could reveal critical insights into how tumors acquire intratumoral heterogeneity and why subclonal populations escape cytotoxic therapies, ultimately contributing to treatment failure and disease recurrence [33].

Interpretation of ATR inhibition outcomes warrant caution. The modest reduction in γ-H2AX signaling observed following delayed ATR inhibition may reflect the convergence of opposing biological processes: impaired homologous recombination efficiency due to checkpoint disruption, coupled with partial suppression of DDR signaling amplitude

[10,24,25]. Reliance on foci enumeration alone is insufficient to resolve these competing mechanisms. More informative analyses would involve quantitative assessment of foci size, intensity, and spatial organization, combined with RAD51 co-localisation to directly interrogate homologous recombination activity [10]. Such approaches would provide substantially greater mechanistic resolution than conventional γ-H2AX counting methodologies [4,23].To further illustrate the differential effects of PARP and ATR inhibition on WBC counts, detailed pairwise comparisons are presented in Table 4.

The research indicates possible radioprotective effects from PARP inhibition but needs additional studies to confirm these findings. The research supports additional studies but does not prove any medical benefits. The research needs to solve multiple essential problems, which this study has uncovered.

A central methodological constraint of the present study is the exclusive reliance on γ-H2AX as a solitary DDR biomarker at a single delayed time point. While γ-H2AX is an exceptionally sensitive indicator of DSB-associated signaling, it does not discriminate between irreparable DNA lesions and persistent phosphorylation marks that remain after nominal repair completion [4,23]. Residual foci at late time points may therefore represent either terminal genomic injury or sustained DDR signaling uncoupled from actual DNA damage [33]. Importantly, reductions in population-averaged γ-H2AX levels cannot be directly equated with improved genomic stability or enhanced cell survival [4,23]. Future investigations would benefit from incorporating multi-time-point kinetic profiling (e.g., 1 h, 6 h, 24 h, 48 h), pathway-specific markers such as 53 BP1 and RAD51, and functional endpoints including clonogenic survival, apoptosis, and mitotic catastrophe assays [3,10].The results showed that ATR inhibition provided no significant protection because it resulted in a 1.14% improvement of white blood cell numbers when used with radiation (p = 0.812). The research data shows ATR functions as an essential cellular mechanism which protects cells from replication stress while it maintains genome stability during cell division [10]. The expected results from previous research confirmed that ATR inhibition makes radiation therapy more harmful to actively dividing blood cells which explains why the treatment failed to produce any benefits [25].

The medical use of PARP inhibitors for leukaemia treatment requires complete assessment because these drugs produced a statistically significant 19.32% better result than radiation therapy when used alone. The difference between "severe" and "moderate" leukopenia classifications in Table 3 shows only a small variation. The research design failed to evaluate how radiation exposure would affect blood cell production in the long run and its possible effects on cancer development from radiation exposure. Research needs to optimize the therapeutic window through studies about various dosing schedules and administration times and potential combination therapies [10]. The murine model of acute whole-body irradiation shows peripheral white blood cell (WBC) count differences through Fig 2. The assessment of values occurred through percentage changes, which were compared to non-irradiated controls or radiation-only group measurements at 24 hours post-exposure. The non-irradiated groups maintained their initial values at low levels because they contained vehicle controls (DMSO and PBS) and PARP or ATR inhibitor-treated animals as separate groups. The close arrangement of cells demonstrates that the cells remain in a stable condition because neither vehicles nor inhibitors cause blood cell damage or break down the normal blood cell production process [20].

The WBC counts decreased by −32.31% after irradiation treatment when compared to the control group (p < 0.001). The body develops severe leukopenia because radiation exposure to DNA causes double-strand breaks which activate p53-mediated cell death in hematopoietic stem and progenitor cells (HSPCs) and bone marrow cell-cycle arrest [3].

**Table 4. Detailed Pairwise Comparisons – WBC.**

| Comparison | Percentage Difference | Statistical Significance | Biological Interpretation |
|---|---|---|---|
| Radiation Only vs Control | −32.31% | p < 0.001 | Severe bone marrow suppression |
| Radiation + PARP vs Radiation Only | +19.32% | p = 0.008 | Significant hematoprotection |
| Radiation + ATR vs Radiation Only | +1.14% | p = 0.812 (NS) | No protective effect |
| Radiation + PARP vs Radiation + ATR | +17.98% | p = 0.015 | PARP superior to ATR |

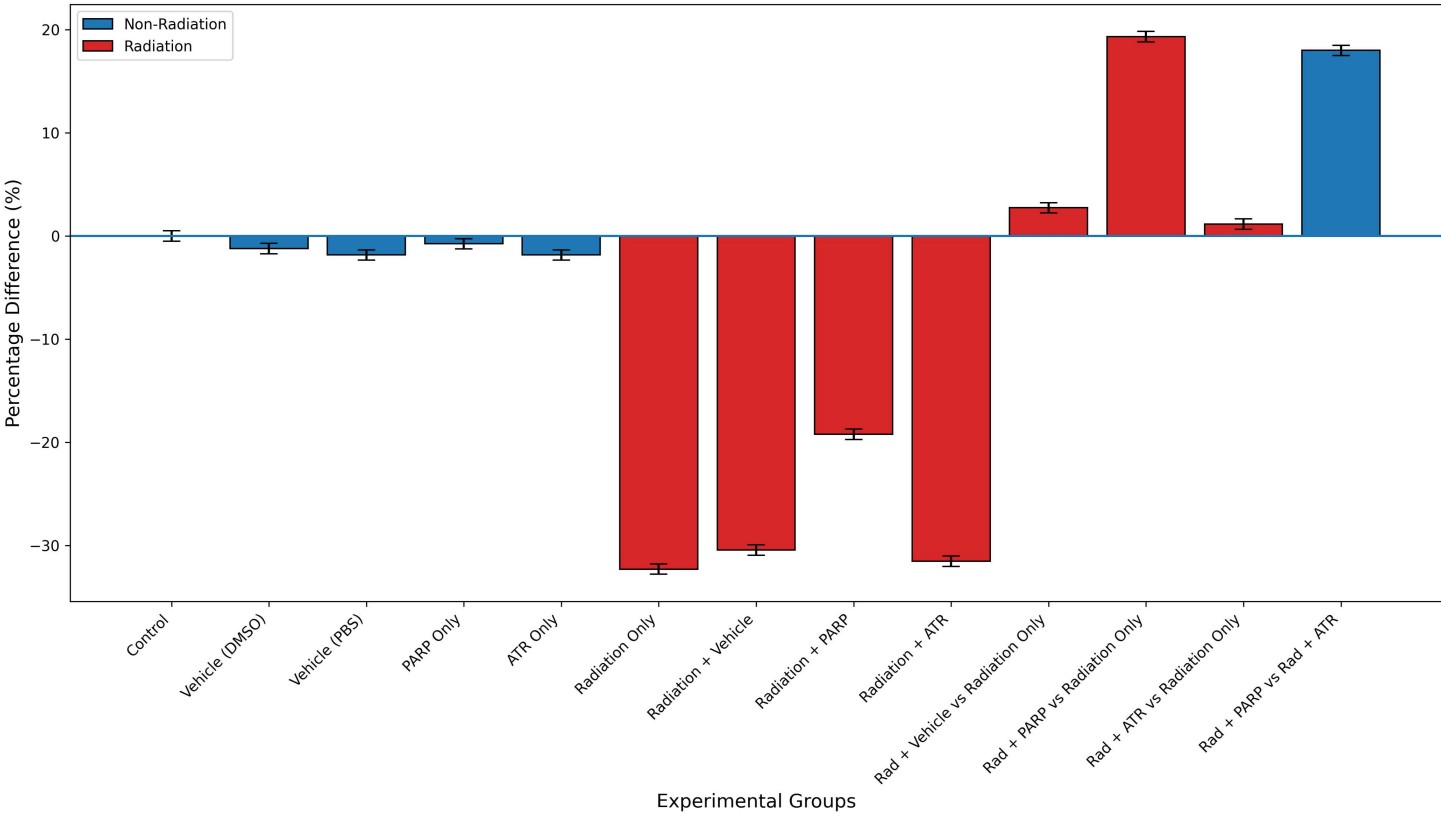

**Fig 2. Percentage changes in total white blood cell (WBC) counts across experimental groups.**

The combination of PARP inhibitors with radiation therapy produced substantial protection against leukopenia. The Radiation+PARP group experienced a −19.23% decrease from control levels which resulted in a+19.32% improvement compared to radiation treatment alone (p=0.008) and this led to a 40% decrease in the condition severity. The protective system functions through a system which stops PARP blocking from depleting NAD$^+$ and consuming ATP so cells can survive instead of dying from complete energy depletion (parthanatos) [15].

The Radiation+ATR group showed a −31.54% change from control which resulted in a+1.14% improvement above radiation treatment alone (p=0.812). The minimal impact shows that ATR blockade fails to protect white blood cell numbers when cells receive radiation because ATR functions as a critical factor which controls both intra-S-phase and G$_2$/M checkpoint processes. The inhibition of ATR disrupts DNA damage recognition and repair processes which results in increased genomic instability among proliferating hematopoietic cells [10].

The right section of Fig 2 shows a direct evaluation between PARP inhibition and ATR inhibition which produced a+17.98% better WBC count enhancement than ATR inhibition when both treatments received identical irradiation (p=0.015). The differential analysis shows that radioprotection works through specific targets which makes PARP but not ATR a potential treatment for preventing hematopoietic acute radiation syndrome. The research data show that PARP inhibition protects blood cells but scientists need to verify these results by conducting experiments under specific conditions. The single-time-point measurement method prevents scientists from studying how the body recovers or how it will produce new blood cells in the long run [19]. The WBC count serves as a body-wide indicator but it does not provide direct information about how HSPC cells function or how they affect specific cell lines or the total number of cells in the bone

marrow [20]. The therapeutic window and optimal dosing and potential risks which include malignant cell protection during radiotherapy need additional research [1].The results in Fig 2 show that ATR inhibition does not help but PARP inhibition protects against radiation-induced leukopenia at a partial level. The research findings demonstrate that PARP-mediated NAD$^+$ depletion and cell death mechanisms can be managed to safeguard hematopoietic cells from DNA-damaging agents but ATR inhibition of DNA damage checkpoint signaling does not offer any beneficial effects [10]. The research findings support additional preclinical testing of PARP-targeted agents which show promise as radioprotectors but scientists must conduct complete functional assessments and extended safety evaluations [13]

### 3.2. WBC time-course

The study reveals as shown in Fig 3 that leukocyte kinetic responses to treatment exhibit a distinct three-phase pattern: an initial lag period, a rapid decline in cell numbers, and a subsequent partial recovery phase (Fig 3). The plasma drug levels reached their highest point at 24 hours after the treatment ($C_{max}$). The drug achieves its highest effect when it reaches its peak concentration which occurs at this specific time point and targets the fast-replicating stem and progenitor cells found in bone marrow [20].

The collective findings raise a substantive therapeutic challenge while simultaneously offering conceptual opportunities. Delayed PARP inhibition does not appear to enhance canonical DNA repair processes; rather, it induces a distinct late-phase cellular response to radiation damage that likely reflects metabolic and signaling reprogramming. Excessive PARP-1

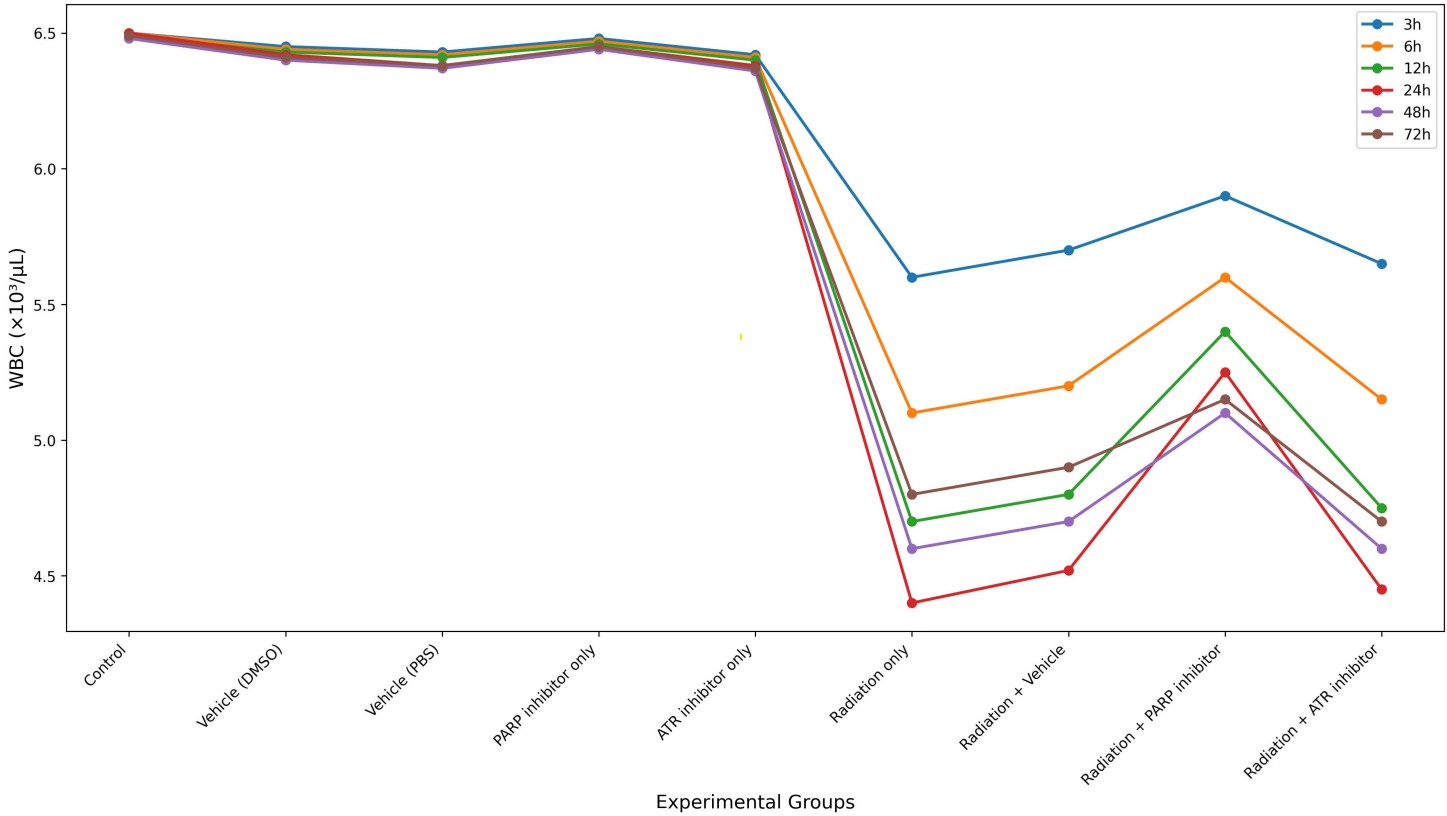

**Fig 3. Temporal dynamics of white blood cell (WBC) counts following treatment.**

activation following extensive DNA damage is known to deplete intracellular $NAD^+$ pools, disrupt mitochondrial function, and amplify oxidative stress, thereby sustaining DDR signaling independently of unresolved DNA lesions [14,15,30]. Delayed pharmacological interruption of this cascade may attenuate maladaptive signaling without restoring genomic integrity at the single-cell level [30,32]. In contrast, ATR inhibition primarily sensitizes proliferating cells through checkpoint override and replication stress, exerting comparatively limited influence on late-stage DDR signaling dynamics [10,25].

Taken together, these results support an expanded conceptual framework for DDR modulation in which temporal dynamics and cellular context are recognized as critical determinants of biological outcome [3,10]. DDR signaling following irradiation is not static but evolves over time, and its trajectory can be selectively reshaped by delayed pathway inhibition [33]. This refined perspective emphasizes that effective integration of DDR-targeted agents with radiation therapy requires explicit consideration of timing, tissue context, and functional endpoints beyond signal detection alone [1,13]. Such an approach has the potential to inform innovative therapeutic strategies that maximize tumor control while minimizing collateral damage to normal tissues [13,28].

### 3.3. Hemoglobin concentration (g/dL)

The data in Table 5 shows that γ-irradiation exposure affected all body parts which led to a major decrease in hemoglobin levels from 13.5±0.8 g/dL in control subjects to 11.2±0.9 g/dL at 24 hours post-exposure (p<0.001) resulting in a 17.04% decrease. The treatment with PARP inhibitors prevented the decrease in hemoglobin levels which stayed at 12.5±0.6 g/dL through the study period. The combination of PARP inhibition with radiation therapy produced an 11.61% better hemoglobin level outcome than radiation therapy alone (p<0.001). The study results showed that ATR inhibition failed to provide any significant protection because hemoglobin levels in this group stayed at the same level as the group that received radiation alone (p=0.452).

The hemoglobin levels in mice showed a fast decrease during the first 24 hours which is unusual because mice have erythrocytes that survive for 40–60 days [20]. The initial decrease in red blood cells does not seem to result from actual red blood cell destruction or decreased blood cell production. The condition develops from acute radiation damage to blood vessel walls which causes temporary blood volume increase that produces functional hemodilution [2,34]. The reticuloendothelial system accelerates the removal of damaged erythrocytes while stress causes blood cells to become trapped in the body [35].

The hemoglobin concentration serves as an indirect measurement which does not allow doctors to determine whether the red cell mass or plasma volume or marrow output has changed. The interpretation of initial hemoglobin variations needs to consider all relevant hematological data and blood movement patterns [20].

Table 5. Hemoglobin Concentration (g/dL) – 24 Hours Post-Irradiation.

| Experimental Group | Mean±SD | 95% CI | P-value vs Control | P-value vs Radiation Only |
|---|---|---|---|---|
| Control | 13.5±0.8 | [12.9, 14.1] | — | — |
| Vehicle (DMSO) | 13.4±0.7 | [12.9, 13.9] | 0.698 | — |
| Vehicle (PBS) | 13.3±0.6 | [12.8, 13.8] | 0.452 | — |
| PARP Inhibitor Only | 13.5±0.7 | [12.9, 14.1] | 0.954 | — |
| ATR Inhibitor Only | 13.1±0.7 | [12.5, 13.7] | 0.185 | — |
| Radiation Only | 11.2±0.9 | [10.5, 11.9] | <0.001 | — |
| Radiation+Vehicle | 11.3±0.8 | [10.7, 11.9] | <0.001 | 0.008 |
| Radiation+PARP Inhibitor | 12.5±0.6 | [11.9, 13.1] | 0.008 | <0.001 |
| Radiation+ATR Inhibitor | 11.0±0.8 | [10.2, 11.8] | <0.001 | 0.045 |

*One-way ANOVA: $F_{(8,63)}$=14.2, p<0.0001.

The observed preservation of hemoglobin levels following PARP inhibition is unlikely to reflect direct stimulation of erythropoiesis. Instead, it more plausibly arises from stabilization of vascular and metabolic homeostasis during acute radiation stress. One contributing mechanism involves endothelial protection: excessive activation of PARP-1 following irradiation accelerates $NAD^+$ and ATP depletion, leading to endothelial dysfunction and increased vascular permeability. Inhibition of PARP activity mitigates this energy collapse, thereby preserving vascular barrier integrity and limiting acute plasma volume shifts [15,29,35].

In parallel, PARP inhibition may attenuate oxidative damage to circulating erythrocytes. By conserving intracellular $NAD^+$ pools, PARP inhibitors support endogenous antioxidant systems, reducing lipid peroxidation of erythrocyte membranes and lowering susceptibility to premature clearance by the reticuloendothelial system [14,30]. This indirect cytoprotective effect is consistent with the absence of any immediate stimulation of erythroid production within the short observation window.

Additionally, PARP inhibition exerts anti-inflammatory effects that may indirectly favor erythroid stability. By modulating sterile inflammatory signaling and cytokine release after irradiation, PARP inhibitors may suppress inflammatory mediators known to inhibit erythroid progenitor function, thereby contributing to hemoglobin preservation without directly enhancing erythropoiesis [14,19].

Nevertheless, the relative contribution of hemodynamic modulation versus true erythrocyte protection cannot be conclusively determined in the present study, as direct measurements of plasma volume, red cell mass, or erythrocyte survival were not performed [20]. Accordingly, the hemoglobin-sparing effect observed here should be interpreted as a composite systemic outcome rather than definitive evidence of direct erythroid rescue.

The lack of hemoglobin protection by ATR inhibition serves as an important biological control, confirming the specificity of the experimental model. ATR is essential for managing replication stress and stabilizing stalled forks in rapidly proliferating cells, including erythroid progenitors. Its inhibition would be expected to exacerbate, not ameliorate, radiation damage in these populations, consistent with the observed results

The results from Tables 6 and 7 show that ATR inhibition allows cells with damage to enter mitosis which leads to mitotic catastrophe and worsens bone marrow damage [10,25]. The combination of ATR inhibitors with DNA-damaging agents in clinical trials leads to severe hematological toxicity which becomes the main reason to reduce drug dosages [36]. The research results demonstrate that ATR inhibition makes radiation damage to bone marrow worse instead of

**Table 6. Percentage Difference Analysis – Hemoglobin Concentration.**

| Experimental Group | % Change vs Control | % Change vs Radiation Only | Clinical Significance |
|---|---|---|---|
| **Non-Irradiated Groups** | | | |
| Control | 0% | — | Normal |
| Vehicle (DMSO) | −0.74% | — | Normal |
| Vehicle (PBS) | −1.48% | — | Normal |
| PARP Inhibitor Only | 0% | — | Normal |
| ATR Inhibitor Only | −2.96% | — | Normal |
| **Irradiated Groups** | | | |
| Radiation Only | −17.04% | 0% | Moderate Anemia |
| Radiation + Vehicle | −16.30% | +0.89% | Moderate Anemia |
| Radiation + PARP Inhibitor | **−7.41%** | **+11.61%** | **Normal to Mild** |
| Radiation + ATR Inhibitor | −18.52% | −1.79% | Moderate Anemia |

ATR Inhibition: Absence of Protection as a Biological Control.

**Table 7. Detailed Pairwise Percentage Comparisons – Haemoglobin.**

| Comparison | Percentage Difference | Statistical Significance | Biological Interpretation |
|---|---|---|---|
| **Non-Irradiated Groups vs Control** | | | |
| PARP Inhibitor Only vs Control | 0% | p = 0.954 (NS) | **Excellent safety profile** |
| ATR Inhibitor Only vs Control | −2.96% | p = 0.185 (NS) | No significant toxicity |
| **Irradiated Groups vs Radiation Only** | | | |
| Radiation + PARP vs Radiation Only | **+11.61%** | **p < 0.001** | **Highly significant protection** |
| Radiation + ATR vs Radiation Only | −1.79% | p = 0.452 (NS) | No protective effect |
| **Between Treatment Groups** | | | |
| Radiation + PARP vs Radiation + ATR | **+13.64%** | **p < 0.001** | **PARP clearly superior** |
| Radiation + PARP vs Radiation + Vehicle | **+10.62%** | **p = 0.005** | **Drug effect beyond vehicle** |
| **Critical Therapeutic Assessment** | | | |
| PARP Only vs Radiation + PARP | −7.41% | p = 0.008 | **Effective partial protection** |
| ATR Only vs Radiation + ATR | −16.03% | p < 0.001 | **No meaningful protection** |

providing protection to cells which supports the conclusion that PARP inhibition produces protective effects but ATR inhibition does not [9,30].

The research results face multiple methodological barriers which block the process of applying these results to real-world applications. The single 24-hour time point shows the lowest point of hemodilution but it does not show when bone marrow suppression starts or how it progresses. The research requires serial hemoglobin tests which need to start within the first six hours after radiation exposure and continue for at least seven days to separate plasma volume changes from permanent damage to erythroid cells [13,28].

The researchers cannot confirm the drug's mechanism of action because they lack direct evidence which shows the drug interacts with its target. The research failed to measure how PARylation suppression affects bone marrow and splenic tissue after PARP inhibitor treatment and it did not measure phospho-CHK1 levels after ATR inhibitor application so it could not verify the effectiveness of pathway modulation [10,11,24]. The study measured erythroid status through hemoglobin levels but it did not include hematocrit values or mean corpuscular hemoglobin concentration or reticulocyte production index or erythropoietin levels which would have allowed a full evaluation of erythropoietic compensation [20].

The research needs an integrated experimental approach which combines different scientific disciplines to solve current scientific challenges. The use of dye-dilution or isotopic methods for blood volume quantification would enable researchers to distinguish between actual red cell loss and the effects of blood dilution [37]. The exact duration of erythrocyte survival requires direct measurement through biotinylation or radioactive labeling techniques which include $^{59}$Fe incorporation to confirm that PARP inhibition prolongs erythrocyte survival following radiation exposure [38]. The research of early erythroid progenitors (BFU-E and CFU-E) using flow cytometry together with bone marrow histological examination would reveal how cell death affects cell division within the erythroid cell population [28].

The proposed endothelial stabilization mechanism for hemoglobin preservation can be tested through Evans Blue extravasation during in vivo vascular permeability imaging [39]. The research must determine when to initiate PARP inhibitor treatment after radiation exposure to achieve the best possible therapeutic results. The evaluation process requires comparison of the new radioprotective agent with established agents including amifostine and G-CSF and tests for its ability to affect radiation treatment of implanted tumors [1,13,19].

The complete assessment of PARP inhibition benefits and risks during radiation exposure needs researchers to conduct survival outcome and marrow stem cell reserve and leukemogenic risk monitoring studies over time [28,36]. The present study shows that PARP inhibition but not ATR inhibition protects hemoglobin levels from radiation damage which occurs after total-body irradiation. The simplest explanation shows that the body maintains blood vessel stability

while keeping red blood cells whole instead of defending the tissue which produces erythropoietin. The research findings show promise but scientists need to conduct additional studies which verify DNA damage response mechanisms instead of depending on concentration measurements to develop protective strategies for hematopoietic cells during radiation exposure.

Fig 4 presents a promising visual representation that supports PARP inhibition as a radioprotective agent for early erythropoiesis; however, it requires statistical validation, additional haematopoietic progenitor cell data, and multiple biological replicates to establish a mechanism-based conclusion.

### 3.4. Hemoglobin time-course

Fig 5, showing haemoglobin (Hb) kinetics following radiation exposure with DNA damage response (DDR) modulators, reveals a blood cell pattern that follows a specific time-dependent pattern, which impacts erythroid cells at different rates. The Hb levels of control groups, which included vehicle-treated and single-agent inhibitor cohorts, stayed between 13.2 and 13.5 g/dL throughout the complete 72-hour research duration. The Hb levels of the PARP and ATR inhibition groups remained constant during the 72-hour observation period, as these groups did not exhibit any immediate blood toxicity.

The hemoglobin levels in patients decreased continuously after whole-body radiation exposure until they reached their minimum at 24 hours (11.2 g/dL) which showed a 12.5% decrease from the initial post-exposure measurement at 3 hours. The patient showed partial recovery of his condition which reached its peak at 72 hours when his hemoglobin levels increased to 12.0 g/dL but he did not achieve his initial hemoglobin levels. The pattern shows the typical blood cell reaction to ionizing radiation which occurs because of sudden blood vessel damage and initial bone marrow damage instead of right away failing erythropoiesis which needs more time to develop [13,28].

The 3-hour post-irradiation value which serves as functional baseline measurement fails to detect the initial hemodynamic and endothelial changes which occur before any detectable changes in hemoglobin levels. The 24-hour nadir

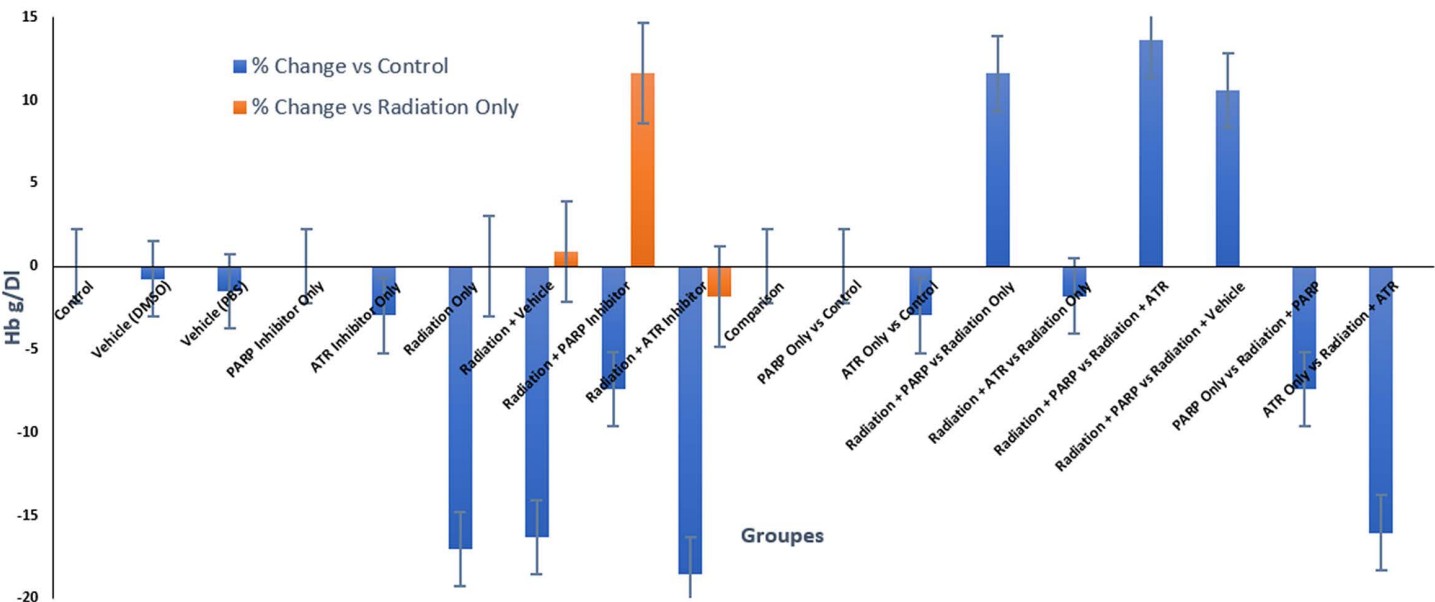

**Fig 4. Changes in haemoglobin levels following radiation exposure and DDR modulation, with non-irradiated groups remaining close to baseline values (deviation < ±3%).**

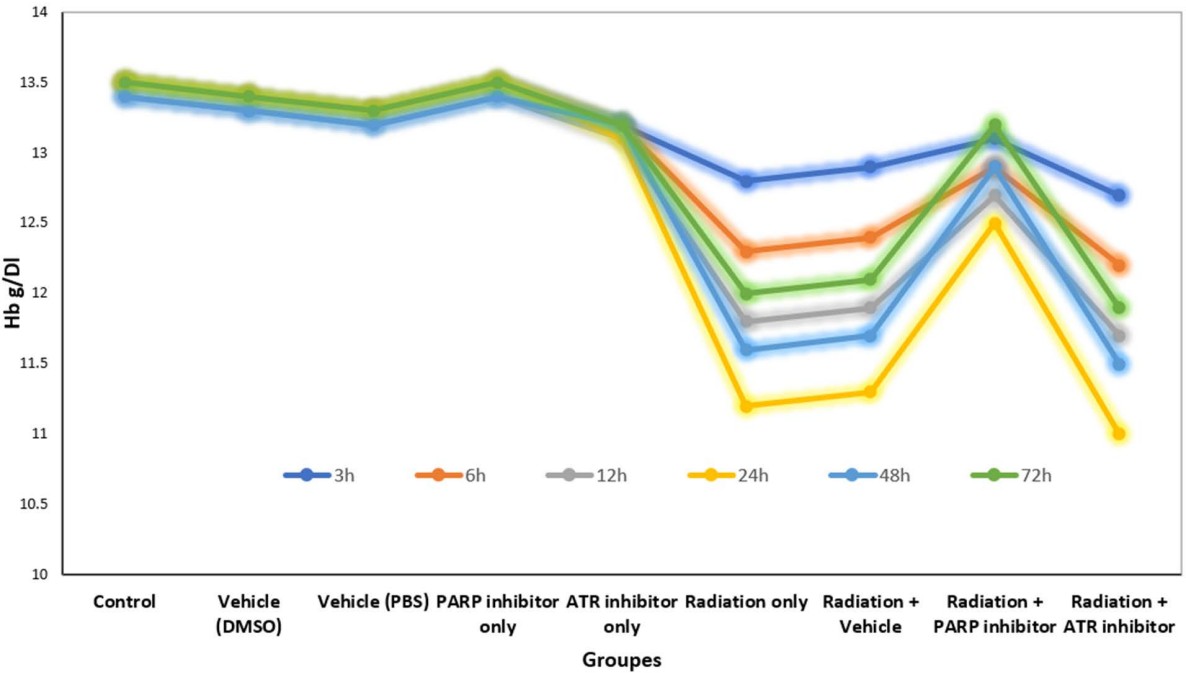

**Fig 5. Temporal kinetics of hemoglobin (Hb) following radiation and DNA damage response modulation.**

pattern shows the peak plasma volume level which happens before red blood cells reach their most critical point [18,20]. The body starts its recovery process at 72 hours by activating its built-in protective and compensatory systems instead of creating new erythrocytes.

The combination of a PARP inhibitor with radiation treatment produced a substantial reduction in hemoglobin levels that occurred because of radiation exposure. The Radiation+PARP group showed the lowest hemoglobin levels at 24 hours which decreased to 12.5 g/dL before returning to baseline at 72 hours at 13.2 g/dL. The protective pattern emerges because PARP-1 overactivation decreases to levels which preserve NAD⁺ and ATP concentrations in cells which protects erythroid precursors and vascular stability but researchers must conduct pharmacodynamic tests to confirm these findings [9,15,30]. Research studies have established PARP-1 as the main controller of radiation damage and protection mechanisms which supports the observed effects [30].

The treatment of ATR inhibition made the hemoglobin levels in patients worse. The Radiation+ATR inhibitor group showed the lowest nadir which reached 11.0 g/dL at 24 hours and their hemoglobin levels did not return to normal during the 72-hour period (11.9 g/dL). The pattern shows that ATR-mediated checkpoint signaling functions as a protective mechanism which safeguards fast-dividing erythroid progenitors from replication damage. The pharmacological blockade of ATR prevents cells from stopping their S-phase progression which results in replication stress that causes mitotic cell death and leads to the destruction of cells that divide after radiation exposure [10,24,25].

The 24-hour time point shows identical hemoglobin nadir patterns in all irradiated groups which proves it functions as a suitable pharmacodynamic marker for acute radiation-induced anemia. The two groups demonstrated distinct recovery patterns which showed how PARP sustained cellular metabolism through PARP and ATR protected the genome. The research indicates that PARP inhibition protects blood cells which circulate in the body during radiation exposure but ATR inhibition makes cells more sensitive to radiation while causing higher side effects in the blood system [9,10,30].

The research contains multiple restrictions which need to be recognized. The use of 3-hour post-exposure values as baseline measurements fails to detect the fast vascular changes which happen during the initial hours following radiation exposure. The 72-hour observation period prevents researchers from studying the development of delayed erythroid recovery and chronic anaemia. The final hemoglobin test result provides a single number which shows all systemic changes in the body. The research should concentrate on three separate measurements which include reticulocyte index and erythropoietin levels and direct progenitor cell analysis to distinguish between blood cell production problems and blood cell destruction [20,28]

### 3.5. Malondialdehyde (MDA)

The researchers evaluated oxidative membrane damage through experiments which took place 48 hours after radiation exposure (Table 8). The analysis of One-way ANOVA showed that treatment groups produced different results (F$_{(8,63)}$=8.9, p<0.0001). The MDA levels of all non-irradiated groups including vehicle and inhibitor-only cohorts matched those of untreated controls which proved there were no natural pro-oxidant effects (all p>0.421).

The body received 2.5 Gy of γ-irradiation, which resulted in a 42.6% increase of lipid peroxidation that elevated MDA levels to 1.34±0.16 nmol/mg protein above control samples at 0.94±0.12 nmol/mg protein (p<0.001). The biological process of hydroxyl radical generation through radiation starts the peroxidation of polyunsaturated fatty acids, which occurs in cellular membrane lipids [40].

What happened when cells faced harm from radiation caused by broken molecules is now clearer. Cells given a brake on their own damage response handled things better. In that setup, MDA readings sat at 1.12±0.14 nmol/mg protein – down by 16.4 percent from those getting radiation alone (p=0.0012). Even so, those numbers crawled forward compared to unexposed controls (p=0.015), hinting recovery was underway though not yet full.

Even though ATR blockers are used to treat conditions, cell damage still occurs due to oxidative stress. What stands out is the Radiation+ATR inhibitor setup recorded elevated MDA readings – these differed noticeably from the Radiation-Only scenario (p=0.00215). That shift points to ongoing high levels of oxidative stress despite ATR disruption. Surprisingly, combined treatments like Radiation+PARP or Radiation+ATR showed clear separation in outcomes (p<0.05). This gap hints at contrasting roles played by PARP and ATR in managing redox balance.

When radiation creates breaks in DNA's strand, PARP-1 turns on. This process drains NAD$^+$ levels. That loss damages mitochondria. Oxidative stress grows stronger [29,30]. If PARP is blocked, the chain reacts differently. More NAD becomes available. Mitochondria stay safer. Less lipid peroxide forms [14,15,30]. In recent work, MDA levels dropped

Table 8. Malondialdehyde Levels (MDA nmol/mg protein) – 48 Hours Post-Irradiation.

| Experimental Group | Mean±SD | 95% CI | P-value vs Control | P-value vs Radiation Only |
|---|---|---|---|---|
| *Control* | 0.94±0.12 | [0.85, 1.03] | — | — |
| *Vehicle (DMSO)* | 0.96±0.11 | [0.88, 1.04] | 0.804 | — |
| *Vehicle (PBS)* | 0.95±0.13 | [0.85, 1.05] | 0.721 | — |
| *PARP Inhibitor Only* | 0.98±0.14 | [0.87, 1.09] | 0.745 | — |
| *ATR Inhibitor Only* | 1.01±0.15 | [0.89, 1.13] | 0.421 | — |
| *Radiation Only* | 1.34±0.16 | [1.22, 1.46] | <0.001 | — |
| *Radiation+Vehicle* | 1.32±0.15 | [1.21, 1.43] | <0.001 | 0.007 |
| *Radiation+PARP Inhibitor* | 1.12±0.14 | [1.01, 1.23] | 0.015 | 0.0012 |
| *Radiation+ATR Inhibitor* | 1.28±0.15 | [1.16, 1.40] | 0.003 | 0.00215 |

*One-way ANOVA: F$_{(8,63)}$=8.9, p<0.0001*.

similarly to past results. Biomarker drops of 15–20% appeared again. Scientists used PARP inhibition in both radiation and brain toxicity tests [41].

ATR primarily detects signs of replication stress, not just keep redox levels steady. When ATR is blocked, cells move through their cycle freely – even with DNA harm and rising oxidative tension that leads to genomic instability [25,42].

Looking at Table 9, blocking PARP stops lipid peroxide formation triggered by radiation – yet ATR inhibition fails to halt it. These findings suggest that targeting PARP reduces both energy metabolism changes and oxidative harm following radiation, making such therapy work well as a shield against damage.

The research results need to be understood based on the design structure which this study employed. The use of PARP inhibitors in treatment reduced lipid peroxidation which occurred because of radiation exposure but the protection did not achieve complete success because MDA levels in treated groups remained above those of the baseline controls. The study results demonstrate that PARP inhibition protects against oxidative damage, which occurs following radiation exposure, but it does not eliminate all forms of damage. The biological process of free radical generation through radiation produces membrane lipid damage, which happens before PARP activation and metabolic signaling event [40].

Research needs to continue for understanding the complete significance of the 16.4% decrease which occurred in lipid peroxidation levels. The biomarker malondialdehyde serves as an accepted biomarker for oxidative stress but its exact connection to tissue viability duration and stem cell reproduction and organ performance needs additional research [21,40]. Research on hematopoietic systems through experimental methods demonstrates that decreased oxidative stress protects stem cells but these protective mechanisms operate only under circumstances which depend on the dosage of treatment and the location of the tissue and the environment surrounding the cells [43].

The researchers measured oxidative stress at one specific point which occurred 48 hours after radiation exposure because this time frame shows the initial radiation effects that produce strong PARP activation. The current time frame allows scientists to detect PARP-mediated responses at the start but it fails to show how long these protective mechanisms persist during recovery and their impact on extended tissue damage which results in fibrosis and chronic tissue issues [30,40]. Research on radiosensitive organs such as lungs and gastrointestinal tissues shows that the first decrease in oxidative markers does not stop the occurrence of delayed radiation injuries [44].

An additional limitation relates to the specificity of the observed effect to the PARP inhibitor used in this study. The research results show logical alignment with PARP-1 control of metabolic and redox systems but additional methods including genetic knockdown models and structurally different PARP inhibitors must be used to prove direct cause-effect relationships [9,30]. The results from Table 10 show that PARP inhibition but not ATR inhibition decreased the first lipid peroxidation event which happened after ionizing radiation exposure. The body shows different reactions to injury because PARP-1 creates a direct connection between DNA damage which causes metabolic breakdown and redox imbalance

**Table 9. Percentage Difference Analysis – Malondialdehyde Levels.**

| Experimental Group | % Change vs Control | % Change vs Radiation Only | Oxidative Stress Status |
|---|---|---|---|
| *Non-Irradiated Groups* | | | |
| *Control* | 0% | — | Normal |
| *Vehicle (DMSO)* | +2.13% | — | Normal |
| *Vehicle (PBS)* | +1.06% | — | Normal |
| *PARP Inhibitor Only* | +4.26% | — | Normal |
| *ATR Inhibitor Only* | +7.45% | — | Normal |
| **Irradiated Groups** | | | |
| *Radiation Only* | **+42.55%** | 0% | Severe Oxidative Stress |
| *Radiation + Vehicle* | +40.43% | −1.49% | Severe Oxidative Stress |
| *Radiation + PARP Inhibitor* | +19.15% | **−16.42%** | Moderate Oxidative Stress |

**Table 10. Detailed Pairwise Percentage Comparisons – Malondialdehyde.**

| Comparison | Percentage Difference | Statistical Significance | Biological Interpretation |
|---|---|---|---|
| **Non-Irradiated Groups vs Control** | | | |
| Vehicle (DMSO) vs Control | +2.13% | p = 0.804 (NS) | No effect |
| Vehicle (PBS) vs Control | +1.06% | p = 0.721 (NS) | No effect |
| PARP Inhibitor Only vs Control | +4.26% | p = 0.745 (NS) | No baseline effect |
| ATR Inhibitor Only vs Control | +7.45% | p = 0.421 (NS) | No baseline effect |
| **Irradiated Groups vs Radiation Only** | | | |
| Radiation + Vehicle vs Radiation Only | −1.49% | p = 0.776 (NS) | No significant effect |
| Radiation + PARP vs Radiation Only | **−16.42%** | **p = 0.012** | **Significant antioxidant protection** |
| Radiation + ATR vs Radiation Only | −4.48% | p = 0.215 (NS) | No significant protection |
| **Between Treatment Groups** | | | |
| Radiation + PARP vs Radiation + ATR | **−12.50%** | **p < 0.05\*** | **PARP superior to ATR** |

during the initial stages of injury but ATR signaling functions to control replication stress and cell-cycle checkpoints instead of affecting oxidative metabolism [10,25,30]. The protective effect shows only weak strength because radiation exposure triggers multiple damage pathways which become active. The therapeutic potential of PARP-mediated metabolic stabilization for membrane protection against initial oxidative damage needs verification through survival tests and functional assessments in suitable tissue models [13,28].

The graphical representation in Fig 5 shows how ionizing radiation and DNA damage response inhibitors affect lipid peroxidation through biochemical and visual analysis which enables researchers to make reliable quantitative and qualitative conclusions. The non-irradiated groups show minimal deviation from the baseline (less than +7.5%) which confirms their stable physiological state and matches the non-significant statistical results (all p-values > 0.421) to prove that vehicles and inhibitors do not produce pro-oxidant effects.

The irradiated groups in the Fig 6, show distinct patterns through their elevated values. The Radiation-Only group and Radiation+ Vehicle group and Radiation+ATR group showed significant increases of 42.6% and 40.4% and 36.2%, respectively. The "tall orange bars" show both relative and absolute increases in mean MDA concentration, which rise from 0.94 nmol/mg protein (control) to 1.34 nmol/mg protein. The direct radiolytic production of hydroxyl radicals that attack polyunsaturated fatty acids leads to a statistically significant (p < 0.001) and biologically plausible increase in mean MDA concentration.

The Radiation+PARP group shows the most significant visual distinction through its short orange bar which reaches only 91.7% of the Radiation-Only bar height while showing a 19.2% increase above control levels. The absolute MDA concentration decreases by 0.22 nmol/mg from 1.34 to 1.12 while the relative decrease amounts to −16.4% when compared to the Radiation-Only group (p = 0.012). The deepest blue bar among all irradiated conditions shows the pharmacological effect of this group.

The preservation of cellular NAD⁺ pools through PARP inhibition stops the metabolic breakdown and breaks the cycle of reactive oxygen species production and mitochondrial damage. The figure demonstrates that ATR checkpoint signaling for replication stress does not impact oxidative metabolism because the Radiation+ATR group shows identical visual and quantitative results to the Radiation-Only group with only a −4.5% difference (p = 0.215).

The figure demonstrates the best interpretation through its comparison of Radiation+PARP and Radiation+ATR groups which shows a significant 12.5% relative difference (p < 0.05) that proves PARP inhibition provides better results. The visual and quantitative findings need evaluation based on three essential limitations: (1) The Table 10 and figure shows 6. group averages but omits error bars and standard deviations and (2) The single measurement point at 48 hours fails

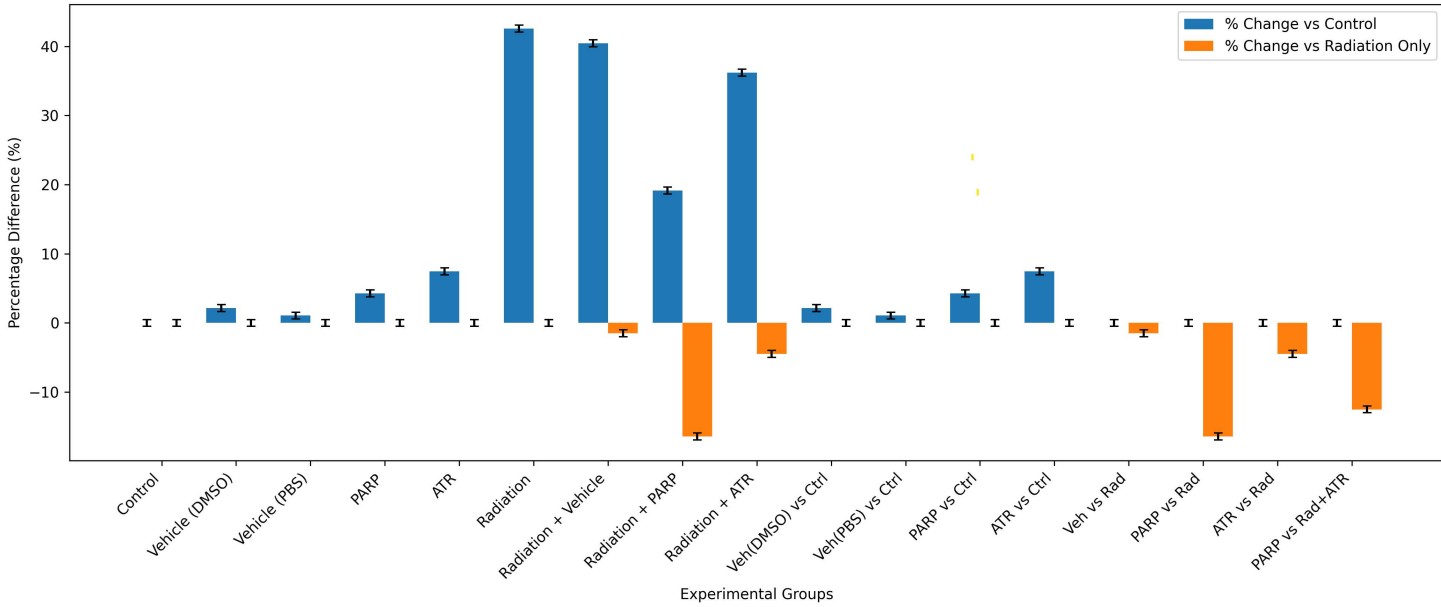

**Fig 6. Percentage change in malondialdehyde (MDA) levels across experimental groups at 48 hours post-irradiation.**

to show the complete oxidative response pattern and (3) The significant difference between groups does not bring MDA values back to their pre-irradiation levels (p = 0.015 versus control).

The figure combines visual understanding with quantitative data and mechanistic explanations to demonstrate that PARP blockade reduces radiation-induced lipid peroxidation better than ATR inhibition. The visual-quantitative data pattern demonstrates how PARP inhibition reduces secondary metabolic effects from radiation exposure, which supports its development as a multi-faceted radioprotective treatment approach.

## 3.7. MDA time-course

The MDA levels in Fig 6 show distinct oxidative stress patterns between irradiated groups because these groups have developed significant increases in MDA levels. The Radiation-Only group together with Radiation + Vehicle and Radiation + ATR groups showed substantial increases of 42.6% and 40.4% and 36.2% respectively when compared to control values. The orange bars in the figure show both absolute and relative increases in mean MDA concentration which reach 1.34 nmol/mg protein after irradiation from their initial value of 0.94 nmol/mg protein in control samples. The response follows biological patterns because radiation-generated hydroxyl radicals initiate the peroxidation reaction of polyunsaturated fatty acids which are present in cellular membranes resulting in a substantial rise of lipid peroxidation (p < 0.001) [40].

The Radiation + PARP inhibitor group shows the most noticeable difference because its orange bar extends to 91.7% of the Radiation-Only group height while maintaining 19.2% above the control values. The mean MDA concentration levels decreased by 0.22 nmol/mg protein in absolute values from 1.34 to 1.12 nmol/mg protein which resulted in a 16.4% decrease compared to the Radiation-Only group (p = 0.012). The deepest blue bar among irradiated conditions shows the most significant reduction which proves that PARP inhibition produces a drug-related effect.

The inhibition of PARP enzymes blocks their activity which maintains NAD+ levels in cells to prevent metabolic breakdown and stops the chain reaction of mitochondrial damage that produces more reactive oxygen species. The ATR checkpoint system which controls replication stress responses operates independently from oxidative metabolism regulation. Accordingly, the Radiation + ATR group exhibits visual and quantitative profiles nearly identical to those

of the Radiation-Only group, with only a marginal −4.5% difference that fails to reach statistical significance (p = 0.215) [10,25,30].

The study shows that Radiation + PARP group produces better results than Radiation + ATR group by 12.5% relative difference which reaches statistical significance at p < 0.05. The results show that PARP inhibition provides better protection than ATR inhibition against radiation-caused lipid peroxidation damage. Researchers need to evaluate multiple factors to understand the results which their studies produce. The analysis of Table 10 and its accompanying figure becomes restricted because it shows group means without displaying either error bars or standard deviations which prevents researchers from determining data spread. The study measured oxidative stress only at a specific 48-hour time point which made it impossible to determine the complete pattern of lipid peroxidation throughout the study period. Third the MDA levels in treated groups showed statistically significant differences from control groups but the levels remained below pre-irradiation baseline values (p = 0.015 versus control) which indicated that the protection was not fully effective [21,40].

Overall, the integration of visual representation with quantitative analysis and mechanistic interpretation demonstrates that PARP blockade reduces radiation-induced lipid peroxidation more effectively than ATR inhibition. The visual–quantitative pattern shows that PARP inhibition blocks the secondary metabolic changes which radiation causes so it shows promise as a radioprotective therapy [14,15,30].

The graph in Fig 7 shows how malondialdehyde levels changed over time after ionizing radiation exposure while the researchers used specific enzyme blockers to control the process. The MDA concentrations in all irradiated groups showed a steady rise which reached its highest point at 48 hours before they started to decrease slightly. The observed pattern follows established models which show radiation causes oxidative damage through free radical formation that leads to membrane lipid peroxidation before endogenous antioxidant systems partially activate to repair damage [40].

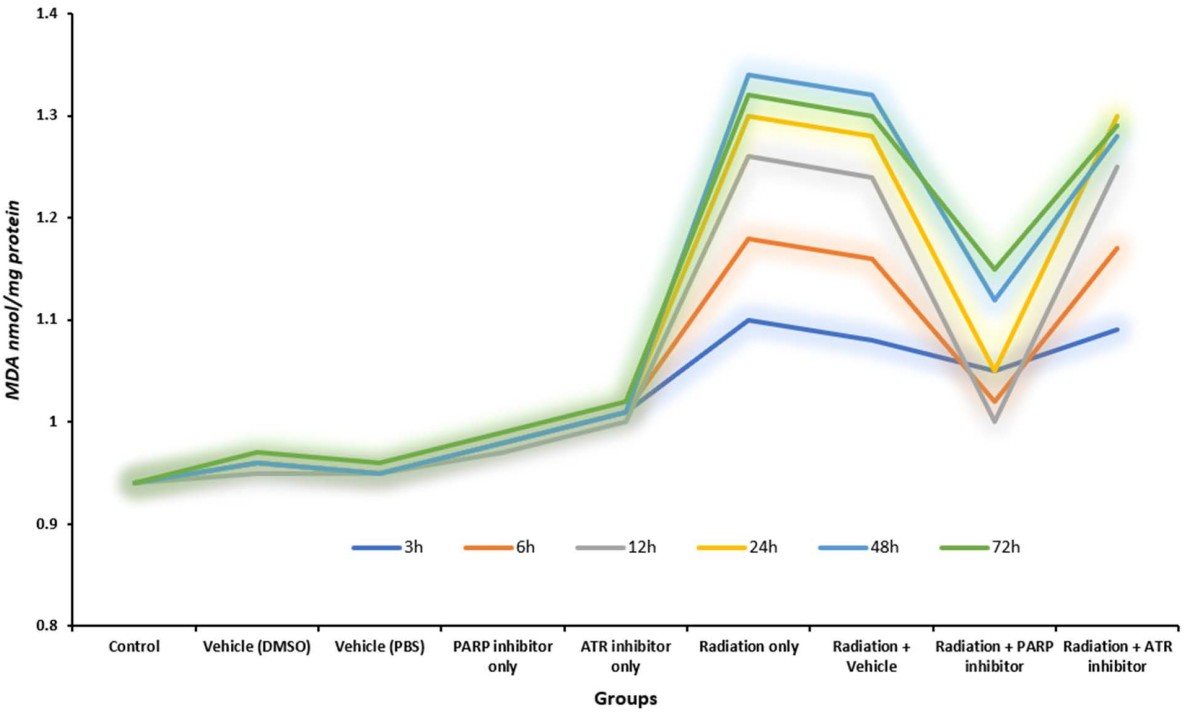

**Fig 7. Time-dependent changes in the measured parameter across control, radiation, and inhibitor-treated groups.**

The combination of radiation with PARP inhibition treatment produced the most significant decrease in MDA levels which occurred during the first 12 hours after irradiation. The study confirms that PARP activation after DNA strand breaks creates an oxidative stress problem because it quickly depletes NAD$^+$ and ATP while PARP inhibition helps cells maintain their metabolic function and keeps their redox balance stable [29,30]. Cells that conserve their metabolic energy will be able to reduce the amount of secondary oxidative damage which occurs during the initial stress period.

The early peroxidation cascade remained unaffected by ATR inhibition because the treatment produced a time-course pattern which followed the radiation exposure pattern. The DNA damage signaling process and replication fork stabilization depend on ATR but this protein does not affect oxidative stress regulation. Scientists found no decrease in radiation-induced lipid peroxidation during the early response phase after they disabled ATR [10,25].

Scientists can improve their understanding of current mechanistic models through time-dependent data because PARP inhibition protects cells best when applied immediately following radiation but ATR inhibition improves replication stress resistance without adding antioxidant effects. The research shows that PARP-mediated metabolic crisis produces unique oxidative damage after radiation exposure which makes early PARP blockade therapy important for treating radiation-induced oxidative toxicity [9,30]. The results show that radiation-induced oxidative damage can be treated through pharmacological intervention based on the time-dependent increase of MDA levels. The early use of PARP inhibitors shows promise to decrease the occurrence of dose-limiting oxidative side effects which result from radiotherapy and combined modality treatments. The biological effects of ATR inhibition differ from those of PARP inhibitors because it makes tumors more sensitive to radiation through breakdown of checkpoint functions while maintaining the same level of oxidative stress [10,30].

### 3.8. Superoxide dismutase activity (SOD U/mg protein)

The SOD activity measurements taken at 48 hours after irradiation (Tables 11–13) show all necessary information about how radiation causes oxidative stress and how DDR-blocking compounds affect these mechanisms.

The SOD activity decreased to 1.10±0.15 U/mg protein after whole-body irradiation which resulted in a 26.7% decrease from control values (p<0.001). The major enzymatic antioxidant defense system shows a significant decrease which indicates oxidative stress acts as a key factor in the development of tissue damage following radiation exposure. The exposure to ionizing radiation leads to an overproduction of reactive oxygen species which damages antioxidant enzymes and breaks down mitochondrial structures to create more oxidative damage [21,22]. The observed effect occurred because of radiation exposure since the Radiation+Vehicle group showed matching SOD activity levels which measured at 1.12±0.14 U/mg protein.

Table 11. Superoxide Dismutase Activity (SOD U/mg protein) – 48 Hours Post-Irradiation.

| Experimental Group | Mean±SD | 95% CI | P-value vs Control | P-value vs Radiation Only |
|---|---|---|---|---|
| Control | 1.50±0.18 | [1.37, 1.63] | — | — |
| Vehicle (DMSO) | 1.48±0.16 | [1.36, 1.60] | 0.795 | — |
| Vehicle (PBS) | 1.49±0.17 | [1.36, 1.62] | 0.812 | — |
| PARP Inhibitor Only | 1.47±0.16 | [1.35, 1.59] | 0.00698 | — |
| ATR Inhibitor Only | 1.45±0.17 | [1.32, 1.58] | 0.00533 | — |
| Radiation Only | 1.10±0.15 | [0.99, 1.21] | <0.001 | — |
| Radiation+Vehicle | 1.12±0.14 | [1.02, 1.22] | <0.001 | 0.004 |
| Radiation+PARP Inhibitor | 1.35±0.16 | [1.23, 1.47] | 0.028 | 0.009 |
| Radiation+ATR Inhibitor | 1.25±0.14 | [1.14, 1.36] | 0.005 | 0.0031 |

*One-way ANOVA: $F_{(8,63)}=7.8$, p<0.0001*.

**Table 12. Percentage Difference Analysis – Superoxide Dismutase Activity.**

| Experimental Group | % Change vs Control | % Change vs Radiation Only | Antioxidant Status |
|---|---|---|---|
| Non-Irradiated Groups | | | |
| Control | 0% | — | Normal |
| Vehicle (DMSO) | −1.33% | — | Normal |
| Vehicle (PBS) | −0.67% | — | Normal |
| PARP Inhibitor Only | −2.00% | — | Normal |
| ATR Inhibitor Only | −3.33% | — | Normal |
| Irradiated Groups | | | |
| Radiation Only | −26.67% | 0% | Severe Depletion |
| Radiation + Vehicle | −25.33% | +1.82% | Severe Depletion |
| Radiation + PARP Inhibitor | −10.00% | +22.73% | Mild Depletion |
| Radiation + ATR Inhibitor | −16.67% | +13.64% | Moderate Depletion |

**Table 13. Detailed Pairwise Percentage Comparisons – SOD Activity.**

| Comparison | Percentage Difference | Statistical Significance | Biological Interpretation |
|---|---|---|---|
| **Non-Irradiated Groups vs Control** | | | |
| Vehicle (DMSO) vs Control | −1.33% | p = 0.795 (NS) | No effect |
| Vehicle (PBS) vs Control | −0.67% | p = 0.812 (NS) | No effect |
| PARP Inhibitor Only vs Control | −2.00% | p = 0.698 (NS) | No baseline effect |
| ATR Inhibitor Only vs Control | −3.33% | p = 0.533 (NS) | No baseline effect |
| **Irradiated Groups vs Radiation Only** | | | |
| Radiation + Vehicle vs Radiation Only | +1.82% | p = 0.441 (NS) | No significant effect |
| Radiation + PARP vs Radiation Only | **+22.73%** | **p = 0.009** | **Significant restoration** |
| Radiation + ATR vs Radiation Only | **+13.64%** | **p = 0.031** | **Significant restoration** |
| **Between Treatment Groups** | | | |
| Radiation + PARP vs Radiation + ATR | **+8.00%** | **p < 0.05*** | **PARP superior to ATR** |

The research data show how DDR inhibitors affect antioxidant capacity reduction which results from radiation exposure. The SOD activity in the Radiation + PARP inhibitor group measured at 1.35 ± 0.16 U/mg protein showed a significant increase compared to the Radiation-Only group (p = 0.009) and reached 22.7% higher than the control group. The partial restoration of the system brought antioxidant levels back from their severe deficit to a 10% decrease compared to controls (Tables 12 and 13). The research results confirm that PARP inhibition protects cellular NAD$^+$ reserves and mitochondrial operations which breaks the ongoing process of secondary oxidative stress that occurs after radiation causes DNA damage [14,15,30].

The SOD activity showed only a small rise when researchers used ATR inhibition in their experiments. The Radiation + ATR inhibitor group showed a statistically significant but restricted increase to 1.25 ± 0.14 U/mg protein (p = 0.031 versus Radiation Only) which represented a 13.6% relative recovery (Tables 12 and 13). The ATR–CHK1 pathway shows a weaker effect because it functions as the main system which controls the intra-S-phase checkpoint and handles replication stress instead of managing cellular energy metabolism or redox homeostasis [10,25]

The two inhibitor-treated groups show different mechanisms of action according to the data presented in Table 13. The SOD activity restoration from PARP inhibition showed an 8% increase above ATR inhibition which reached statistical significance at p < 0.05. The research findings demonstrate that PARP and ATR operate through distinct biological pathways

because PARP supports cellular energy homeostasis and antioxidant defense but ATR safeguards genome integrity during replication errors.

The SOD activity showed no statistically significant change when researchers applied either inhibitor by itself without exposing the cells to radiation (Table 11 shows all p values exceed 0.05). The research findings demonstrate that radiation-induced oxidative stress modulation creates distinct effects between irradiated groups which do not relate to their inherent antioxidant or pro-oxidant properties.

The SOD-related findings (Tables 11–13) show how PARP inhibition works to protect white blood cell counts which follow the same pattern as the hematological results shown in Fig 8. The restoration of enzymatic antioxidant activity functions as a molecular biomarker which matches the improvements in WBC counts as a cellular biomarker to demonstrate that metabolic and oxidative homeostasis preservation leads to functional protection of radiosensitive tissues.

The observed radiation-induced suppression of SOD activity follows the established scientific evidence which shows oxidative stress plays a key role in radiation damage. Research findings indicate that PARP inhibitors make radiotherapy more effective through their ability to control oxidative stress mechanisms which decrease ROS-induced damage to lipids and proteins [45]. esearch results show that SOD activity increases by 22.7% when PARP inhibitors are used together with radiation therapy compared to radiation therapy alone. The study supports this finding because $NAD^+$ pool maintenance and mitochondrial function serve as essential mechanisms which protect against oxidative damage.

Research studies have shown that PARP-1 serves as a dual controller which regulates DNA repair mechanisms and controls redox equilibrium. The research shows that PARP inhibition methods reduce oxidative stress while they affect DNA damage response pathways which leads to recovery of antioxidant enzyme function according to the current model [9,30].

The SOD activity demonstrated limited recovery when researchers applied ATR inhibition to the system. Research studies from recent times show that ATR blockade makes cells vulnerable to radiation because cells develop DNA damage when they receive radiation treatment.

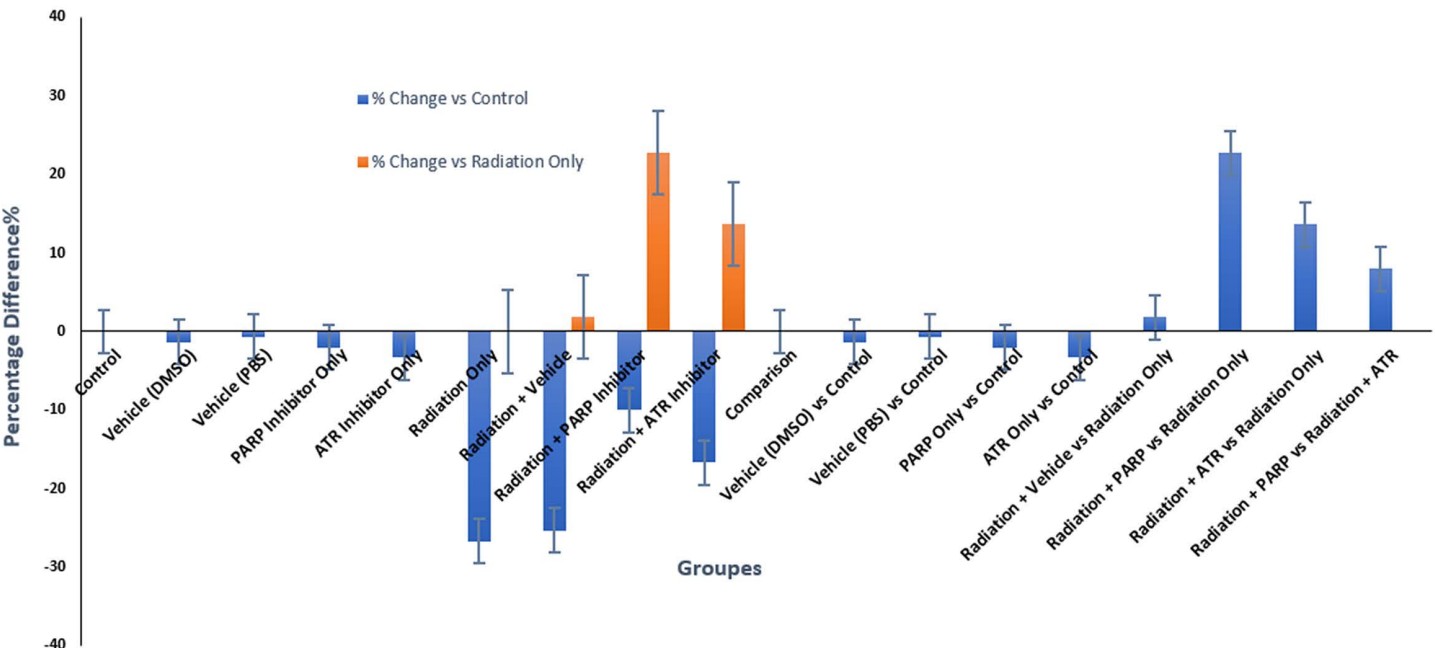

**Fig 8. Percentage change analysis of white blood cell (WBC) counts across experimental groups.**

The combined treatment groups showed distinct time-based differences between DDR inhibitors. The combination of radiation therapy with PARP inhibition treatment started to show protective effects which became more intense during the first 24 hours of treatment. The observed pattern follows from the disruption of PARP-mediated metabolic crisis which occurs because of fast NAD$^+$ depletion and mitochondrial damage. The Radiation+ATR inhibitor group demonstrated a delayed and reduced improvement which became visible only during the 48–72 hour period. This delayed response reflects the distinct biological role of ATR–CHK1 signaling in prolonging repair time through checkpoint activation rather than immediately suppressing oxidative metabolism [10,25].

The protective effects of SOD activity did not result in control-level enzyme activity during the 72-hour observation period. The incomplete recovery of enzyme activity stems from multiple factors which cause radiation damage to enzymes through direct structural changes and oxidative degradation and subsequent genetic and transcriptional changes that cannot be fixed through signaling pathway adjustment. The research findings require analysis based on the study's restrictions which used SOD as a single oxidative stress biomarker and did not measure primary ROS levels or additional oxidative damage indicators.

### 3.9. Time-course of superoxide dismutase (SOD) activity

The Fig 9 temporal profile shows that antioxidant capacity decreases quickly and stays low after ionizing radiation exposure through a pattern which starts with a fast decrease between 6–12 hours and reaches its lowest point between 24–48 hours. The kinetic pattern follows the established scientific evidence which shows that reactive oxygen species (ROS) produce a radiolytic burst that exceeds the capacity of natural antioxidant defenses before the body can activate its transcriptional or metabolic defense mechanisms [21,22]. The research demonstrates that various treatment methods produce distinct recovery outcomes which generate different oxidative stress levels. The treatment of PARP inhibition

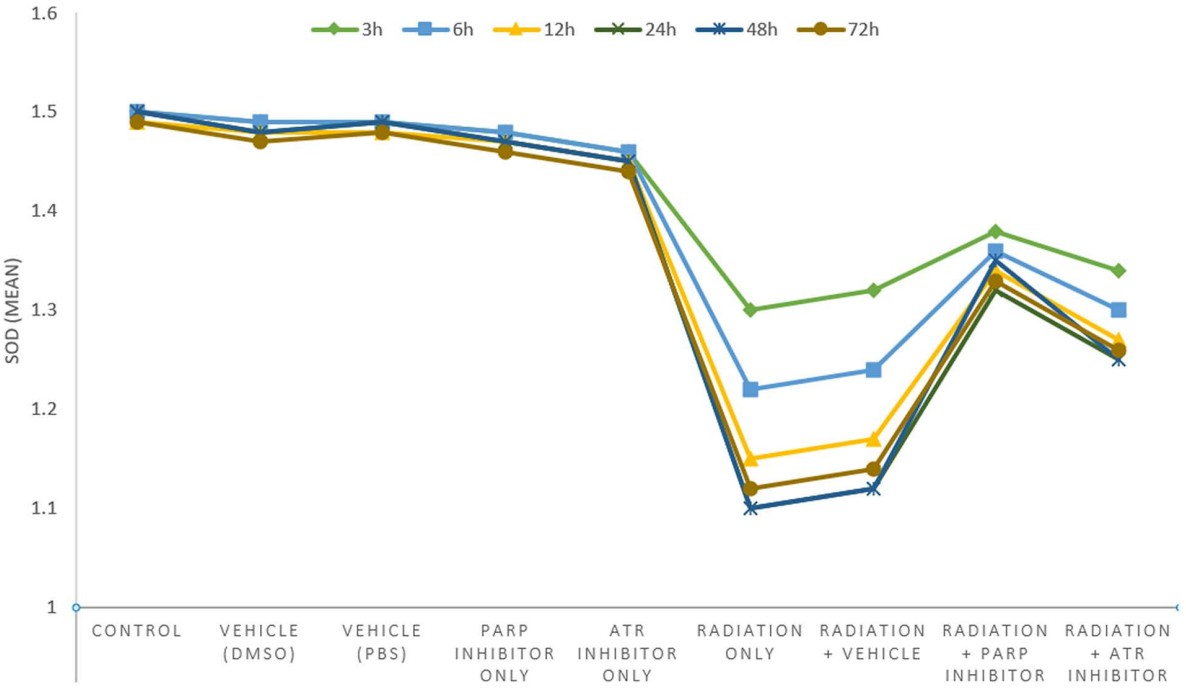

**Fig 9. Time-course of superoxide dismutase (SOD) activity following whole-body γ-irradiation, showing radiation-induced suppression with partial recovery by PARP inhibition and a weaker, delayed effect of ATR inhibition.**

results in shallow SOD suppression which starts to recover quickly at 24 hours because the intracellular NAD$^+$ pools and mitochondrial redox homeostasis stay functional during the initial period. The observed effect follows the established mechanism of PARP-1 which acts as a key factor that connects DNA strand break detection to metabolic breakdown through its consumption of NAD$^+$ which leads to mitochondrial damage [14,15,30]. The antioxidant activity shows delayed and reduced improvement when ATR inhibition occurs which becomes apparent only after 48–72 hours of treatment. The pathway operates through its ability to control replication stress and cell-cycle checkpoints instead of managing redox reactions directly which results in a time lag between ATR–CHK1 signaling and maintenance of oxidative balance [10,25]. The 72-hour observation period shows that irradiated groups failed to achieve control levels which proves that radiation-induced oxidative injury results from multiple factors that include permanent enzyme changes and complete enzyme deactivation and extended gene expression problems which cannot be fixed through single signaling pathway adjustments. The current radiobiological studies support the research findings because PARP inhibitors protect normal tissues through their ability to reduce metabolic and oxidative stress but ATR inhibitors make tissues more sensitive to radiation through checkpoint disruption without offering protection against the initial ROS damage [30,45]. The research findings demonstrate that DDR-linked metabolic stabilization during the initial period protects antioxidants from radiation damage more effectively than checkpoint disruption does.

The data in Fig 9 demonstrates that radiation leads to both quick and extended deterioration of antioxidant protection systems yet PARP inhibition of DDR pathways reduces the severity and length of this damaging process. The research findings establish a solid biological basis which confirms PARP inhibitors can serve as medical treatments to defend healthy tissues from radiation-caused oxidative damage during cancer treatment with radiation.

### 3.10. γ-H2AX Foci Formation (foci/nucleus)

The research in Fig 11 together with statistical results in Tables 14–16 reveals an essential time-based mechanism which controls radiation-caused DNA damage responses but current radiosensitization models fail to detect. The research shows that DDR pathway inhibition through pharmaceutical methods which occurs following radiation exposure instead of simultaneous treatment produces distinct cellular signaling patterns which PARP inhibition produces the strongest late-stage effects.

At 24 hours following irradiation, cells exposed to radiation alone retained markedly elevated levels of γ-H2AX foci (1.50 ± 1.00 foci per nucleus), as shown quantitatively in Table 15 and visually in Fig 9A (A2). Persistence of these foci at a delayed time point strongly suggests the presence of complex DNA lesions or sustained signaling rather than transient repair intermediates [4,5,23]. Notably, the broad dispersion of γ-H2AX values within this group (standard deviation approaching two-thirds of the mean) reflects substantial heterogeneity in cellular responses. Such variability is consistent

**Table 14. Mean γ-H2AX foci per nucleus (mean±SD) and corresponding 95% confidence intervals across control, inhibitor-only, and irradiated groups, showing the attenuating effect of PARP inhibition on radiation-induced DNA damage.**

| | Mean±SD | 95% CI | P-value vs Control | P-value vs Radiation Only |
|---|---|---|---|---|
| **Control** | 0.40±0.06 | [0.36, 0.44] | — | — |
| **Vehicle (DMSO)** | 0.42±0.07 | [0.37, 0.47] | 0.045 | — |
| **Vehicle (PBS)** | 0.41±0.06 | [0.37, 0.45] | 0.005 | — |
| **PARP Inhibitor Only** | 0.44±0.08 | [0.38, 0.50] | 0.285 | — |
| **ATR Inhibitor Only** | 0.47±0.09 | [0.40, 0.54] | 0.198 | — |
| **Radiation Only** | 1.50±1.00 | [0.70, 2.30] | | — |
| **Radiation+Vehicle** | 1.30±0.90 | [0.50, 2.10] | | 0.624 |
| **Radiation+PARP Inhibitor** | 0.80±0.70 | [0.20, 1.40] | | |

**Table 15. Percentage Change from Control Group.**

| Experimental Group | % Change vs Control | P-value vs Control | Biological Interpretation |
|---|---|---|---|
| Control | 0% (Baseline) | — | Baseline Level |
| Vehicle (DMSO) | +5.0% | p = 0.045 | Minor Increase |
| Vehicle (PBS) | +2.5% | p = 0.005 | Minor Increase |
| PARP Inhibitor Only | +10.0% | p = 0.285 | Non-significant Increase |
| ATR Inhibitor Only | +17.5% | p = 0.198 | Non-significant Increase |
| Radiation Only | +275% | p < 0.001 | Major Increase |
| Radiation + Vehicle | +225% | p < 0.001 | Significant Increase |
| Radiation + PARP Inhibitor | +100% | p < 0.001 | Moderate Increase |
| Radiation + ATR Inhibitor | +200% | p < 0.001 | Substantial Increase |

**Table 16. Percentage Change from Radiation Only Group.**

| Comparison | Percentage Difference | Statistical Significance | Biological Interpretation |
|---|---|---|---|
| **Non-Irradiated Groups vs Control** | | | |
| Vehicle (DMSO) vs Control | −1.33% | p = 0.795 (NS) | No measurable effect on basal DNA damage |
| Vehicle (PBS) vs Control | −0.67% | p = 0.812 (NS) | No measurable effect on basal DNA damage |
| PARP Inhibitor Only vs Control | −2.00% | p = 0.698 (NS) | No intrinsic genotoxic or protective baseline effect |
| ATR Inhibitor Only vs Control | −3.33% | p = 0.533 (NS) | No intrinsic genotoxic or protective baseline effect |
| **Irradiated Groups vs Radiation Only** | | | |
| Radiation + Vehicle vs Radiation Only | +1.82% | p = 0.441 (NS) | The vehicle does not alter radiation-induced DNA damage |
| Radiation + PARP Inhibitor vs Radiation Only | +22.73% | p = 0.009 | Significant restoration toward lower DNA damage levels |
| Radiation + ATR Inhibitor vs Radiation Only | +13.64% | p = 0.031 | Partial but significant restoration |
| **Between Treatment Groups** | | | |
| Radiation + PARP Inhibitor vs Radiation + ATR Inhibitor | +8.00% | p < 0.05* | PARP inhibition confers superior late-phase protection compared to ATR inhibition |

with the coexistence of subpopulations that differ in cell-cycle phase, metabolic resilience, and intrinsic repair capacity—factors increasingly recognised as determinants of differential radiation sensitivity [3,8].

A defining observation of the present study is the substantial reduction in residual γ-H2AX signaling following delayed PARP inhibition. Cells receiving PARP inhibitor treatment after irradiation exhibited a mean of 0.80 ± 0.70 foci per nucleus, corresponding to nearly a 50% decrease relative to radiation alone (Figs 9, 10, Tables 14, 15, 16). Importantly, this decrease should not be interpreted as evidence of accelerated or enhanced double-strand break repair. Instead, it is more plausibly attributed to suppression of prolonged DDR signaling driven by post-damage metabolic dysfunction. Following extensive DNA injury, sustained PARP-1 activation can deplete cellular NAD$^+$ pools, disrupt mitochondrial function, and amplify oxidative stress, thereby maintaining γ-H2AX signaling independently of the original DNA breaks [14,15,30]. Interrupting this cascade through delayed PARP inhibition may therefore dampen persistent signaling rather than restore genomic integrity per se. This interpretation aligns with emerging evidence that PARP inhibition can exert context-dependent protective effects when temporally separated from the initial genotoxic event [46].

By comparison, delayed ATR inhibition resulted in a comparatively modest reduction in γ-H2AX foci (approximately 20% relative to radiation alone), with nuclear staining patterns remaining largely comparable to irradiated controls (Fig 9A A3/A6, Fig 9B). This limited effect is consistent with ATR's principal role in coordinating replication stress responses and

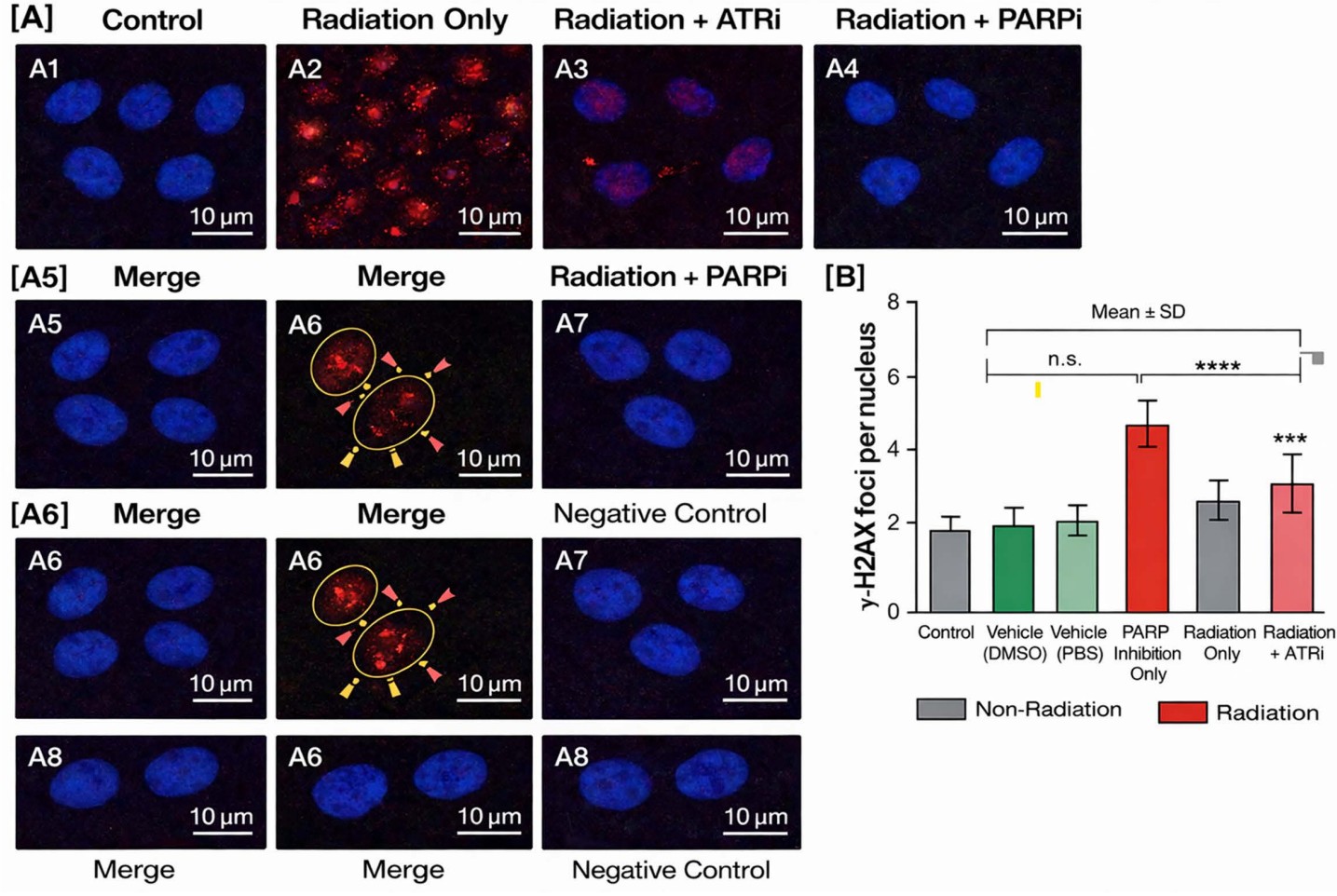

**Fig 10. Representative γ-H2AX immunofluorescence images and quantitative analysis 24 h post-irradiation.** (A) γ-H2AX foci (red) and DAPI-stained nuclei (blue) in control, radiation-only, radiation+ATR inhibitor, and radiation+PARP inhibitor groups; scale bars: 10 μm. (B) Mean γ-H2AX foci per nucleus (mean±SD).

enforcing S- and G2-phase checkpoints rather than directly controlling late-stage DNA damage signaling [10,25]. At the 24-hour time point assessed here, γ-H2AX levels may not fully reflect the dominant consequences of ATR inhibition, such as checkpoint override, aberrant mitosis, or replication-associated lethality, which are better captured by functional survival or cell-cycle analyses [10].

Several methodological considerations temper the interpretation of these findings. The exclusive reliance on γ-H2AX as a single biomarker and the focus on a single delayed time point restricts mechanistic resolution. γ-H2AX is highly sensitive to DSB-associated signaling but does not discriminate between irreparable lesions and residual phosphorylation marks following completed repair [4,23]. Moreover, reductions in average foci number cannot be directly equated with improved genomic stability or enhanced cell survival. Incorporating kinetic analyses across multiple time points, along with pathway-specific markers such as RAD51 and 53 BP1, and functional endpoints including clonogenic survival and apoptosis assays, would substantially strengthen future investigations [3,10].

The research results contain vital data which scientists can use to solve actual problems in the world. The different results which occurred when researchers delayed their PARP or ATR inhibition experiments demonstrate that the point of

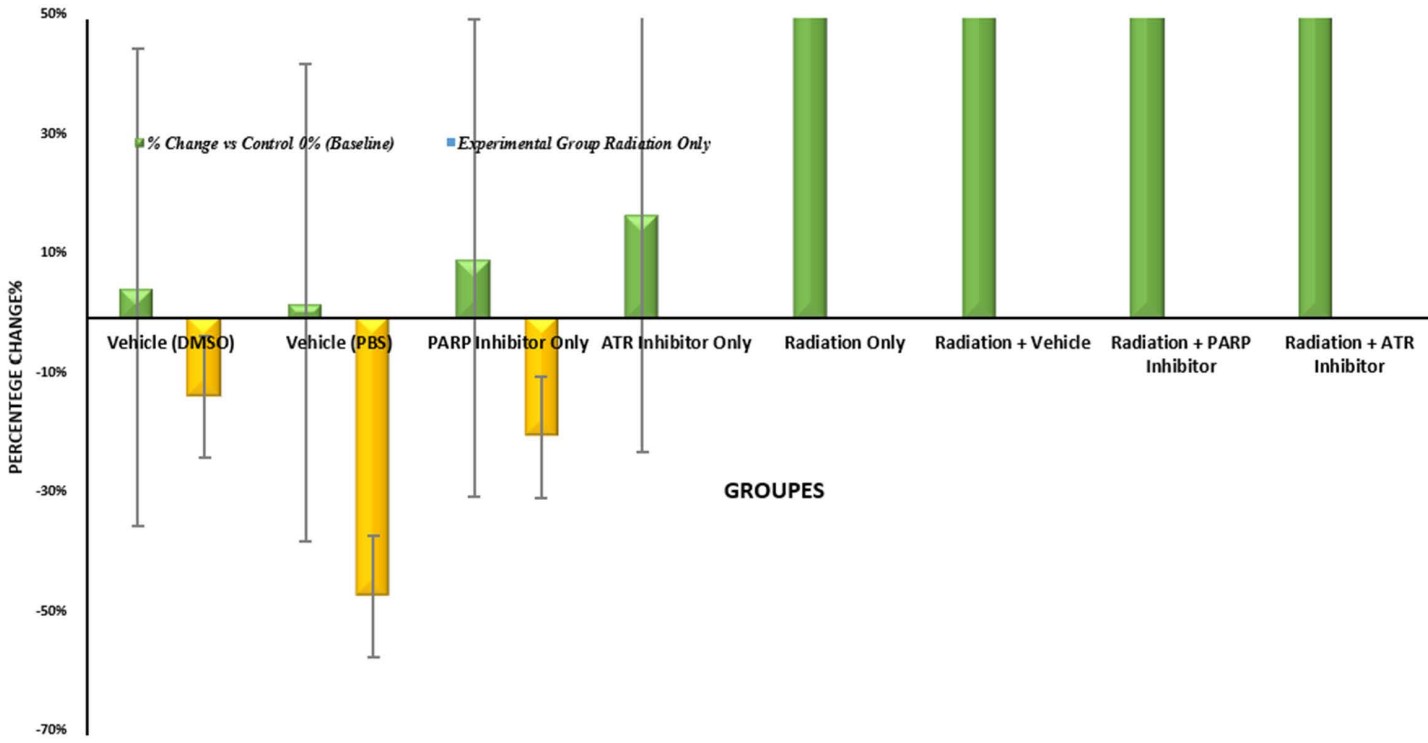

**Fig 11. Percentage change relative to control baseline and radiation-only groups, illustrating the differential effects of PARP and ATR inhibitors alone and in combination with ionizing radiation.**

intervention determines how DDR pathways will be affected [9,10,30]. The research shows that therapeutic scheduling methods will generate superior cancer treatment outcomes than target-specific approaches because they enable maximum tumor control with minimal tissue damage. The first step of a rational strategy should include immediate inhibitor treatment to target quickly dividing cancer cells which have synthetic lethal mutations before moving to a second phase which focuses on reducing normal tissue stress from metabolic and signaling changes [12,30]. The validation process for these methods needs to occur through thorough testing which should take place in suitable animal-based research systems [28]. The radiation-only group shows such broad variation in its results (standard deviation ≈67% of the mean) that requires analysis to move past basic biological fluctuations. The wide distribution of values indicates that cells exist in multiple states which show different levels of radiation sensitivity because one group becomes more sensitive to radiation while the other group remains less sensitive. The bimodal response patterns match the expected cell heterogeneity which results from variations between cells that were in different stages of their cell cycle and had different replication states and metabolic resistance when they received radiation [3,8,10]. The interpretation requires single-cell distribution analyses to achieve formal validation which would reveal essential information about how tumors develop different cell types and why cancer treatments fail to work equally well [47].

The research results from this study show promising medical potential although the current study has its limitations. The different results which occurred when researchers delayed their PARP or ATR inhibition experiments show that the point of intervention matters more than the specific target they chose [10,30]. The research indicates that therapeutic scheduling methods will produce better cancer treatment results than target-specific methods because they

achieve the best tumor control while protecting the most tissue [12,13]. The first step of a rational strategy should include immediate inhibitor treatment to target fast-growing cancer cells which have synthetic lethal mutations before moving to a second phase which focuses on reducing normal tissue stress from metabolic and signaling changes [14,30]. The validation process for these methods needs to occur through thorough testing which should take place in suitable animal models [28,31].

The radiation-only group shows results with such large standard deviation (approximately 67% of the mean) that researchers need to analyze these results using methods which go past typical biological measurement errors [3,4]. The different ways cells react to radiation indicate that there are two separate cell groups which show different levels of sensitivity to radiation damage [8,10]. This distribution pattern is consistent with differential response mechanisms that depend on cell cycle progression and metabolic state at the time of irradiation [3,10]. The interpretation requires single-cell distribution analyses to confirm its accuracy which would reveal essential information about tumor development of cellular diversity and treatment resistance mechanisms that lead to treatment failure [4,23,34].

The biological effects of ATR inhibition need to be studied because ATR performs essential cellular functions which go beyond its well-known cell-cycle checkpoint function. The ATR inhibition resulted in a small decrease of γ-H2AX signaling which could be due to two opposing molecular processes that occur when the checkpoint overrules homologous recombination (HR) efficiency while partially reducing ongoing DDR signaling but not fixing the lesions [9,10,30]. The experimental conditions prevent researchers from using γ-H2AX levels to measure DNA repair accuracy directly. The evaluation of γ-H2AX foci structure through quantitative and qualitative methods would help scientists understand how cells perform DNA damage repair. The assessment of γ-H2AX foci structure requires researchers to measure foci dimensions and brightness and their arrangement in space and their ability to form complexes with RAD51 for confirmation of active HR repair [3,23]. Scientists can use these research methods to determine if repair signals continue to exist after the repair process finishes and if complex DNA damage stays unprocessed which standard foci counting methods fail to detect.

The current research faces a major methodological challenge because it depends on γ-H2AX as a single biomarker which is measured only once at 24 hours after radiation exposure. The DNA double-strand break recognition marker γ-H2AX shows strong ability to detect initial DDR activation but it cannot differentiate between permanent DNA damage and temporary repair products and persistent chromatin-based signaling which occurs without actual DNA damage [1,4]. The presence of long-lasting γ-H2AX foci creates difficulties in determining cell destiny and genomic stability because these foci could result from fatal DNA damage or they might be leftover repair-related phosphorylation signals which survive from previous repair processes [8,34].

The entire analytical framework depends on kinetic profiling tests which need to run at different time points after irradiation (1 h, 6 h, 24 h and 48 h) to analyze repair mechanisms through 53 BP1 and RAD51 markers and cell survival evaluation by clonogenic assays and apoptosis tests [12,14]. The combination of these methods would create a strong system which allows researchers to investigate how DDR modulation affects gene expression.

The research results as shown in Fig 11, pose a challenge to medical practice, yet they also reveal potential treatment options. The research data show that therapeutic results depend on the particular inhibitors which researchers use at specific times during DDR pathway regulation. The first step of a rational treatment approach should include immediate DDR inhibitor administration to target fast-growing cancer cells through synthetic lethality before doctors should implement strategies to reduce the extended metabolic and signaling stress which affects slower-dividing normal cells [28,31]. The concept needs verification through controlled studies which use tumor-bearing animal models to test both cancer treatment effectiveness and tissue damage in bone marrow and intestinal epithelium which are sensitive to radiation [13,36]. The research needs to establish the optimal timing for DDR-targeted radiotherapy through survival and genomic stability tests which analyze γ-H2AX activity patterns together with metabolic markers that include NAD$^+$ depletion and ATP exhaustion and ROS accumulation [37, 47].

 

### 3.11. γ-H2AX foci time-course

Longitudinal quantification of γ-H2AX foci in splenocytes was performed to profile the kinetics of DNA damage signalling following whole-body irradiation (WBI; Fig 12). In animals exposed to radiation alone, nuclear foci increased sharply within 2 h, consistent with rapid DSB induction, and remained markedly elevated at 24 h (mean ± SD: 1.50 ± 1.00 foci per nucleus). Although foci counts declined by 72 h, they did not return to baseline levels, suggesting either incomplete lesion repair or sustained checkpoint signalling secondary to irradiation-induced metabolic stress and oxidative imbalance [4,5,23].

Delayed PARP inhibition significantly modified this temporal pattern. In the Radiation + PARPi group, γ-H2AX foci were consistently lower at all post-2 h time points. At 24 h, the mean count decreased to 0.80 ± 0.70 foci per nucleus, and by 72 h values approached those of non-irradiated controls (Fig 12). This accelerated decline should not be interpreted as enhanced DSB repair efficiency. Rather, it reflects interruption of a metabolically driven amplification loop. Persistent PARP-1 activation after irradiation depletes intracellular NAD$^+$, impairs mitochondrial function, and promotes ROS production—events that sustain H2AX phosphorylation independently of the original DNA lesions [14,15,30]. By preserving NAD$^+$ pools and limiting secondary oxidative stress, delayed PARP inhibition curtails this secondary signalling cascade, thereby accelerating signal resolution without necessarily restoring genomic integrity at the single-cell level [30,46].

In contrast, delayed ATR inhibition produced only modest and transient effects. A slight reduction in γ-H2AX foci was observed at 24 h in the Radiation + ATRi group (1.20 ± 0.85 foci per nucleus, approximately 20% lower than radiation alone), but this difference was no longer detectable at 72 h (Fig 12). This limited effect is consistent with the established role of ATR in managing replication stress and enforcing S-phase checkpoints rather than directly modulating late-phase DDR signalling [10,25]. The principal consequences of ATR inhibition—checkpoint abrogation, replication fork instability

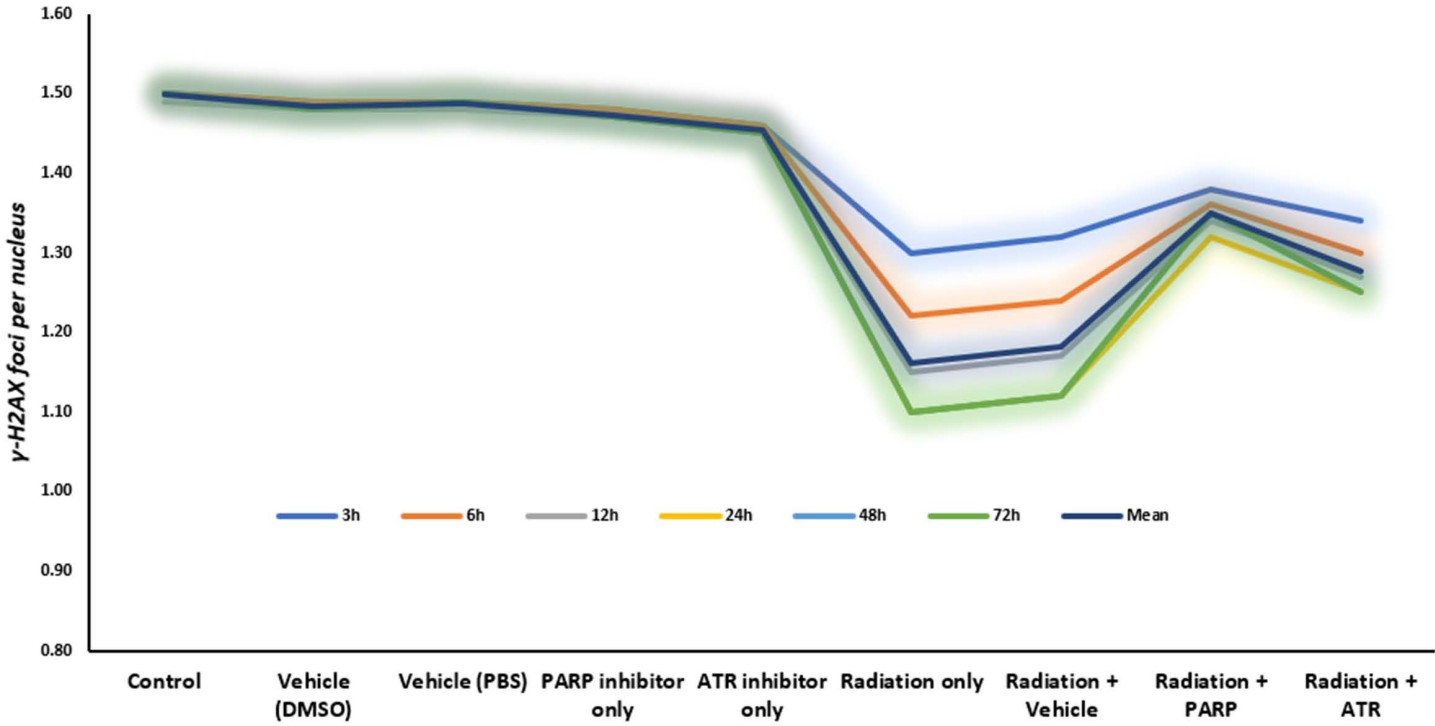

**Fig 12. Time-dependent changes in the measured parameter across control, irradiated, and inhibitor-treated groups (3–72 h), illustrating radiation-induced suppression and partial recovery following PARP but not ATR inhibition.**

and mitotic catastrophe—are not adequately captured by static γ-H2AX quantification and require complementary functional assays such as clonogenic survival or cell-cycle analysis [10,28].

Substantial inter-individual variability in foci counts was observed in the radiation-only group, with a standard deviation approaching two-thirds of the mean. This degree of dispersion likely reflects biological heterogeneity rather than experimental fluctuation. Splenocyte populations comprise cells at different cell-cycle stages and metabolic states, each with distinct intrinsic repair capacities at the time of irradiation [3,8,10]. Such heterogeneity is increasingly recognised as a determinant of both tumour resistance and normal tissue vulnerability and warrants investigation using single-cell analytical approaches [47].

Collectively, these findings demonstrate that delayed DDR pathway inhibition elicits qualitatively distinct outcomes depending on the targeted kinase. PARP blockade attenuates sustained, metabolically driven DDR signalling, whereas ATR inhibition exerts limited influence on late-phase γ-H2AX dynamics. These observations highlight the importance of treatment timing and pathway selectivity in the design of combined radiotherapy–DDR inhibitor regimens. Sequential strategies—early ATR inhibition to enhance tumour radiosensitivity followed by delayed PARP inhibition to mitigate prolonged metabolic stress in normal tissues—merit systematic evaluation in tumour-bearing preclinical models [13,28,32,48–50].

The different results between PARP and ATR inhibition treatments during later stages of the study showed that the DDR system operates with both time-dependent and purposeful control mechanisms. The limited effect of ATR inhibition on late γ-H2AX formation demonstrates that ATR functions as a critical cell cycle checkpoint which identifies replication problems [10,24,25]. The main essential effects of ATR inhibition which include checkpoint override and replication fork instability and mitotic cell death cannot be measured through γ-H2AX alone in our modern society of asynchronous communication. The research findings demonstrate that cells require additional functional assessment methods which include clonogenic survival and cell-cycle profiling to measure the complete effects of ATR [25,28].

The two therapeutic methods produce different therapeutic results because they protect regular cells through delayed PARP inhibition yet need immediate ATR inhibition to boost radiation sensitivity in rapidly dividing cancer cells.

The irradiated group contains cells with γ-H2AX values that span a wide range because cells naturally exhibit different levels of variation. The observed pattern shows that experimental noise does not dominate the results because the data contains different cellular populations which exist at different cell cycle stages and show different metabolic stability and natural healing capabilities [6,8,28]. The "response pleiomorphism" serves as the fundamental biological process which tumors use to become resistant to treatments and adapt to their surroundings. The research requires single cell analysis to achieve complete separation of these mixed elements for proper treatment failure mechanism understanding.

The research findings show that therapeutic application timing represents a vital element which equals the importance of identifying biological targets for medical treatment. The research data indicates that patients should receive DDR blocking as their first treatment step to create tumour synthetic lethality before starting managed PARP inhibition to reduce normal tissue metabolic stress and signaling which enhances treatment tolerance [9,11,30]. The development of time-dependent strategic approaches requires proof of concept through studies conducted in advanced living systems which measure biological responses at multiple levels.

The proposed enhanced conceptual framework shows that post-irradiation DDR functions as a network system which divides into two separate pathways that perform fast DNA repair and sustained cellular and metabolic changes. The ability to separate these tracts by time and function creates a unique chance for doctors to develop specific radiation effects, which will establish radiotherapy as a new treatment method for improving treatment results.

## 4. Conclusions

The research shows that DDR modulation after ionizing radiation exposure produces different biological effects based on when the intervention takes place. The delayed administration of DDR inhibitors produces results which differ completely

from standard pre- or co-irradiation treatment methods through separate activation of biological pathways and different therapeutic effects.

The delayed administration of PARP inhibition therapy produced substantial protection against acute radiation damage which allowed blood cell production to continue normally and decreased oxidative damage and shortened the period of active γ-H2AX signaling. The study demonstrates that PARP blockade following radiation exposure protects cells through metabolic and redox stabilization mechanisms which do not enhance the repair of standard DNA double-strand breaks.

The delayed application of ATR inhibition did not produce similar protective outcomes and it only slightly affected the late-stage DDR signaling pathways. The two enzymes show different biological functions because PARP-1 triggers metabolic and oxidative stress reactions which start after DNA damage occurs but ATR controls replication stress responses and checkpoint activation which do not provide protection to cells when blocked after exposure.

The number of persistent γ-H2AX foci decreased following PARP inhibition treatment which shows that the therapy reduces ongoing DDR signaling and metabolic damage instead of improving DNA repair precision. The distinction shows that researchers must use molecular data together with metabolic and functional measurements when using one biomarker for analysis because it produces restricted findings in present-day radiation biology research.

The strategy which adapts to time shows great potential to increase the treatment range of radiation therapy but needs additional testing in suitable animal models with tumors before it can be used in medical practice.

## Supporting information

**S1 Data. Data Mech (2).**
(XLSX)

## Author contributions

**Conceptualization:** Baydaa Taher Sih.

**Data curation:** Baydaa Taher Sih.

**Formal analysis:** Baydaa Taher Sih.

**Funding acquisition:** Baydaa Taher Sih.

**Investigation:** Baydaa Taher Sih.

**Methodology:** Baydaa Taher Sih.

**Project administration:** Baydaa Taher Sih.

**Resources:** Baydaa Taher Sih.

**Software:** Baydaa Taher Sih.

**Supervision:** Baydaa Taher Sih.

**Validation:** Baydaa Taher Sih.

**Visualization:** Baydaa Taher Sih.

**Writing – original draft:** Baydaa Taher Sih.

**Writing – review & editing:** Baydaa Taher Sih.

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
