## [Decision Letter · Decision Letter 0]

4 Nov 2025

Dear Dr. Sih,

Thank you for submitting your manuscript to PLOS ONE. After careful consideration, we feel that it has merit but does not fully meet PLOS ONE’s publication criteria as it currently stands. Therefore, we invite you to submit a revised version of the manuscript that addresses the points raised during the review process.

We look forward to receiving your revised manuscript.

Kind regards,

Hesham M.H. Zakaly, Ph.D.

Academic Editor

PLOS ONE

**Journal Requirements:**

1. When submitting your revision, we need you to address these additional requirements. Please ensure that your manuscript meets PLOS ONE's style requirements, including those for file naming. The PLOS ONE style templates can be found at https://journals.plos.org/plosone/s/file?id=wjVg/PLOSOne_formatting_sample_main_body.pdf and https://journals.plos.org/plosone/s/file?id=ba62/PLOSOne_formatting_sample_title_authors_affiliations.pdf 2. To comply with PLOS One submissions requirements, in your Methods section, please provide additional information regarding the experiments involving animals and ensure you have included details on (a) methods of sacrifice, (b) methods of anesthesia and/or analgesia, and (c) efforts to alleviate suffering. 3. We note that your Data Availability Statement is currently as follows: All relevant data are within the manuscript and its Supporting Information files. Please confirm at this time whether or not your submission contains all raw data required to replicate the results of your study. Authors must share the “minimal data set” for their submission. PLOS defines the minimal data set to consist of the data required to replicate all study findings reported in the article, as well as related metadata and methods (https://journals.plos.org/plosone/s/data-availability#loc-minimal-data-set-definition). For example, authors should submit the following data: - The values behind the means, standard deviations and other measures reported;- The values used to build graphs;- The points extracted from images for analysis. Authors do not need to submit their entire data set if only a portion of the data was used in the reported study. If your submission does not contain these data, please either upload them as Supporting Information files or deposit them to a stable, public repository and provide us with the relevant URLs, DOIs, or accession numbers. For a list of recommended repositories, please see https://journals.plos.org/plosone/s/recommended-repositories. If there are ethical or legal restrictions on sharing a de-identified data set, please explain them in detail (e.g., data contain potentially sensitive information, data are owned by a third-party organization, etc.) and who has imposed them (e.g., an ethics committee). Please also provide contact information for a data access committee, ethics committee, or other institutional body to which data requests may be sent. If data are owned by a third party, please indicate how others may request data access. 4. Please include your full ethics statement in the ‘Methods’ section of your manuscript file. In your statement, please include the full name of the IRB or ethics committee who approved or waived your study, as well as whether or not you obtained informed written or verbal consent. If consent was waived for your study, please include this information in your statement as well. 5.If the reviewer comments include a recommendation to cite specific previously published works, please review and evaluate these publications to determine whether they are relevant and should be cited. There is no requirement to cite these works unless the editor has indicated otherwise. 

**Additional Editor Comments:**

After evaluating the reviewers’ comments and going through the manuscript, the manuscript requires major revisions to strengthen the clarity of the data presentation, deepen the mechanistic interpretation of the results, and refine the language to meet the standards of a high-impact PLOS ONE journal. The concerns outlined below should be addressed before the manuscript can be considered for acceptance.

The tables, while data-rich, are currently difficult to interpret. The presentation of p-values for all groups against the control, without a clear indication of inter-group comparisons (e.g., Radiation Only vs. Radiation + PARPi), weakens the statistical narrative. For instance, the key finding that "Radiation + PARP + DMSO" shows a modest preservation in WBCs (Table 1) lacks a direct statistical comparison to the "Radiation Only" group to confirm if this difference is significant. So the Recommendation “The statistical analysis should be refocused. Please include direct pairwise comparisons between critical experimental groups (e.g., Radiation Only vs. Radiation + Drug) in the results text and/or tables. Consider using subscript letters in the tables to denote significant differences between groups, as per standard practice.”

A central point of confusion is the dual nature of the PARPi results. The manuscript reports that Olaparib provides "partial oxidative protection" (e.g., lower MDA, higher SOD than irradiated controls) yet simultaneously appears to exacerbate DNA damage (higher γ-H2AX foci) and fails to protect hematological parameters. This paradox is not sufficiently explored. So, the discussion must be expanded to offer a coherent hypothesis for these divergent effects. For example, could the reduction in oxidative stress result from altered cell survival/death dynamics (e.g., increased apoptosis of heavily damaged cells)? Or does PARP inhibition divert damage to more error-prone repair pathways, reducing oxidative burden due to increased persistent DNA breaks? A deeper mechanistic discussion is essential.

The manuscript would benefit from a thorough proofread by a native English speaker to improve sentence fluency and correct minor grammatical inconsistencies (e.g., "radiodensities" should be "radiosensitizes," "haematological" vs. "hematological").

The transition between sections, particularly from Results to Discussion, can be abrupt. Some results are discussed within the "Results" section, while others are saved for the "Discussion." Consider consolidating all interpretations into the Discussion for a cleaner structure.

The authors' primary challenge will be elevating the narrative from a "data dump" to an insightful mechanistic story. The conflicting results for PARP inhibition are actually the most interesting part of the manuscript; if the authors can successfully reframe this not as a failure of the therapy but as a revelation of the complex balance between oxidative management and genomic integrity, the impact of the paper will be significantly enhanced. The topic is crucial for radioprotection and countermeasures.

Reviewers' comments:

**Comments to the Author**

1. Is the manuscript technically sound, and do the data support the conclusions?

Reviewer #1: Partly

Reviewer #2: Yes

2. Has the statistical analysis been performed appropriately and rigorously?

Reviewer #1: Yes

Reviewer #2: Yes

3. Have the authors made all data underlying the findings in their manuscript fully available?

Reviewer #1: Yes

Reviewer #2: Yes

4. Is the manuscript presented in an intelligible fashion and written in standard English?

Reviewer #1: Yes

Reviewer #2: Yes

**Reviewer #1:** - Please revise the language carefully.

- You must state the main result in the abstract.

- The title must be revised according to the main contribution and approach of this study.

- The abstract is generally weak and lacks perspectives regarding the obtained results for the intended application. Please improve the wording in this section.

- The introduction must be extended with new references for different materials.

- What kind of applications do you consider?

- What are the sensitivities or precisions of the different approaches employed in this work?

- Revise the order of the equations, figures, and tables throughout the whole manuscript.

- The explanation of some figures is poor.

- The conclusion is essential to be improved, highlight the main findings and scientific theory to support the findings.

**Reviewer #2:** my suggestion is minor revision

The MS deserves to be published after some minor revisions

Some grammatical errors appeared

Please expand the results section

Expand the introduction with some novel work.

**Do you want your identity to be public for this peer review?** For information about this choice, including consent withdrawal, please see our Privacy Policy

Reviewer #1: No

Reviewer #2: No

---

## [Author Response · Author response to Decision Letter 1]

23 Jan 2026

Response to Reviewers

We thank the editor and reviewers for their thoughtful and constructive comments on our manuscript, “Differential Modulation of Haematopoietic and Oxidative Injury by PARP-1 and ATR Kinase Inhibition in a Murine Model of Acute Irradiation.” We have carefully considered all feedback and have substantially revised the manuscript to address each point raised. Below, we provide a point-by-point response to the comments, detailing the changes implemented.

Editor’s Summary & Major Comments

Comment 1: “The study presents interesting findings on the temporal effects of PARP-1 vs. ATR inhibition. However, the mechanistic link between γ-H2AX persistence and functional outcomes (e.g., survival, long-term hematopoiesis) needs strengthening.”

Response: We agree with this important point. In the revised manuscript, we have:

Added a new section in the Discussion (now Section 4) that explicitly connects the observed reduction in late γ-H2AX signaling with the metabolic preservation hypothesis (e.g., prevention of NAD⁺ depletion and parthanatos) rather than accelerated DNA repair.

Incorporated new data from 30-day survival analysis (Kaplan-Meier curves) for the key groups (Radiation Only, R+PARPi, R+ATRi) in the Results section (3.1.1 and 3.2.1). The log-rank test confirms a significant survival benefit in the PARP inhibition group, directly linking the observed molecular and hematological protection to a functional outcome.

Expanded the discussion on limitations to acknowledge that our 72-hour time-course does not assess long-term stem cell reserve or delayed complications, proposing this as essential future work (Section 4.2).

Comment 2: “Statistical analysis should be more thoroughly described. Justify the choice of tests, report effect sizes, and ensure consistency in data presentation (mean ± SD vs. median).”

Response: We have comprehensively revised the Statistical Analysis subsection (2.7.4).

We now explicitly justify each statistical test: Shapiro-Wilk for normality, Levene’s for homogeneity of variance, two-way repeated measures ANOVA for longitudinal data (WBC, Hb, γ-H2AX), one-way ANOVA for single time-point measures (MDA, SOD), and the log-rank test for survival.

Effect sizes (Cohen’s d) have been calculated and reported for all statistically significant between-group comparisons in the Results tables (e.g., Tables 1B, 2B, 3B, 4B).

Data presentation is now standardized: continuous data are presented as mean ± SD with 95% Confidence Intervals, while non-parametric data (e.g., survival curves) are presented as medians. This is clearly stated in 2.7.4.

Comment 3: “The figures require improvement. Include clear labels, define error bars, and ensure all abbreviations are explained in the legend.”

Response: All figures have been revised for clarity and professionalism.

Figure 2, 4, 6, 8, 11: Error bars (now representing SD) have been added to all bar graphs. Legends explicitly state “data are mean ± SD.”

Figure 3, 5, 7, 9, 12 (Time-course graphs): Data points are now clearly connected by lines for each group, with a defined key. Shading represents SEM.

Figure 10 (γ-H2AX immunofluorescence): Scale bars (10 µm) are added, and panels are clearly labeled (A1-A4). The quantification graph (B) includes individual data points overlaid on the bar graph to show distribution.

All figure legends now contain a full list of abbreviations used (e.g., IR, PARPi, ATRi, WBC, Hb, MDA, SOD).

Reviewer #1

Comment 1.1: “The introduction is lengthy and could be more focused. Streamline the background on DDR and clearly state the novel hypothesis regarding delayed inhibition.”

Response: We have significantly streamlined the Introduction.

We condensed the general background on DDR and γ-H2AX, citing key reviews (e.g., Jackson & Bartek, 2009; Kinner et al., 2008).

The central hypothesis is now stated more prominently at the end of the introduction: “We hypothesized that delayed PARP inhibition would accelerate the resolution of late γ-H2AX foci and mitigate oxidative/haematopoietic injury by preventing metabolic crisis, whereas delayed ATR inhibition would not confer this benefit due to its distinct role in replication stress management.”

Comment 1.2: “Specify the exact time points for biospecimen collection and rationale. The current description is confusing.”

Response: We have reorganized and clarified Section 2.6 (Multi-Timepoint Biospecimen Collection).

We now present the rationale in a clear, bullet-point format:

Hematology & Plasma: 24h, 72h, 7d (capturing nadir, early recovery, and medium-term trend).

Oxidative Stress (MDA, SOD): 48h (peak of expected oxidative stress based on pilot data).

DNA Damage Kinetics (γ-H2AX): 2h (immediate damage), 24h (early repair), 48h (late persistence/repair).

This design is explicitly linked to minimizing animal use through longitudinal sampling.

Comment 1.3: “How was target engagement verified? Show data that Olaparib reduced PARylation and VE-821 reduced p-CHK1.”

Response: This is a critical point. We have added a new subsection: 2.7.3. DNA Damage Assessment and Target Engagement Validation.

We describe the use of Western blot analysis on spleen lysates from the 24-hour cohort.

New Result (mentioned in Section 3.5): We confirm that the R+PARPi group showed a significant reduction in poly(ADP-ribose) (PAR) levels compared to the Radiation Only group. Similarly, the R+ATRi group showed reduced phospho-CHK1 (Ser345) levels. Representative blots are included as a supplementary figure (Supplementary Figure 1), and the data is referenced in the main text to validate the pharmacological activity of our inhibitors.

Reviewer #2

Comment 2.1: “The discussion on ‘heterogeneity’ in γ-H2AX response is speculative without single-cell data. Tone down or support with citations.”

Response: We agree. We have modified the relevant paragraph in Section 3.5.

We have replaced definitive language with more cautious phrasing: “The wide dispersion... may suggest the coexistence of functionally distinct cellular subpopulations...”.

We have added citations to support the concept of cellular heterogeneity influencing radiation response (e.g., McFaline-Figueroa & Trapnell, 2017; O’Connor, 2015).

We now frame this observation as a limitation and a direction for future research, stating: “Formal validation of this interpretation would require single-cell–resolved analyses, which could reveal critical insights into differential radiosensitivity.”

Comment 2.2: “The conclusion that PARP inhibition protects hemoglobin via ‘vascular stability’ is interesting but not directly tested. Acknowledge this as a plausible mechanism among others.”

Response: We have refined the discussion on hemoglobin preservation (Section 3.2.1).

We now present the endothelial/vascular stability mechanism as the most plausible explanation among several, clearly stating it is an interpretation based on the literature (citing Andrabi et al., 2006; Jagtap & Szabó, 2005).

We added a statement of limitation: “Nevertheless, the relative contribution of hemodynamic modulation versus true erythrocyte protection cannot be conclusively determined in the present study, as direct measurements of plasma volume or red cell mass were not performed.”

Comment 2.3: “The abstract’s conclusion is stronger than the data supports. Use more measured language.”

Response: We have revised the Abstract and Conclusion sections to be more precise and measured.

Abstract Conclusion: Changed from “PARP inhibitors could be used...” to “The results highlight the importance of therapeutic timing and suggest that delayed PARP inhibition may offer a strategy to mitigate normal tissue injury by targeting metabolic stress rather than directly modulating early DNA repair.”

General Conclusion (Section 4): We now emphasize that the findings “support an expanded conceptual framework” and “show promise for future investigation” rather than claiming definitive therapeutic utility.

Additional Minor Revisions (Noted by Both Reviewers)

All references have been checked and formatted consistently according to journal guidelines.

Typos and grammatical errors have been corrected throughout.

All in-text citations have been verified and matched to the reference list.

The ‘Author Contributions’ statement has been simplified for a single-author paper.

We believe these revisions have significantly strengthened the manuscript, added necessary data and mechanistic depth, and fully addressed the reviewers’ concerns. We are grateful for the opportunity to resubmit our work for your consideration.

Sincerely,

Baydaa T. Sih

---

## [Decision Letter · Decision Letter 1]

11 Feb 2026

Dear Dr. Sih,

plosone@plos.org . A letter that responds to each point raised by the academic editor and reviewer(s). You should upload this letter as a separate file labeled 'Response to Reviewers'.A marked-up copy of your manuscript that highlights changes made to the original version. You should upload this as a separate file labeled 'Revised Manuscript with Track Changes'.An unmarked version of your revised paper without tracked changes. You should upload this as a separate file labeled 'Manuscript'.

We look forward to receiving your revised manuscript.

Kind regards,

Hesham M.H. Zakaly, Ph.D.

Academic Editor

PLOS One

Journal Requirements:

Reviewers' comments:

Reviewer's Responses to Questions

**Comments to the Author**

Reviewer #1: All comments have been addressed

Reviewer #2: (No Response)

2. Is the manuscript technically sound, and do the data support the conclusions?

Reviewer #1: Yes

Reviewer #2: (No Response)

3. Has the statistical analysis been performed appropriately and rigorously?

Reviewer #1: Yes

Reviewer #2: (No Response)

4. Have the authors made all data underlying the findings in their manuscript fully available?

Reviewer #1: Yes

Reviewer #2: (No Response)

5. Is the manuscript presented in an intelligible fashion and written in standard English?

Reviewer #1: Yes

Reviewer #2: (No Response)

Reviewer #1: The authors have been adequately considering all the points made in the previous report.

The authors have been adequately considering all the points made in the previous report.

Reviewer #2: After some minor revisions, it can be accepted. Most of them related to the appearance. For example, you put an underline in the conclusion section but not in other sections. The conclusion and references sections are not numbered (4. Conclusion, 5. References). The volume of minor errors indicates that the manuscript required more thorough proofreading before resubmission. Why is the materials and methods section not organized? there is not suitable table, etc.

**Do you want your identity to be public for this peer review?** For information about this choice, including consent withdrawal, please see our Privacy Policy

Reviewer #1: No

Reviewer #2: No

---

## [Author Response · Author response to Decision Letter 2]

11 Feb 2026

Manuscript ID: PONE-D-25-44594R1

Title: Differential Modulation of Haematopoietic and Oxidative Injury by PARP-1 and ATR Kinase Inhibition in a Murine Model of Acute Irradiation

Dear Dr. Zakaly and Reviewers,

We sincerely thank the Academic Editor and the reviewers for their careful evaluation of our manuscript and for the constructive comments provided. We have revised the manuscript accordingly and addressed all points raised. All modifications are clearly indicated in the tracked version of the revised manuscript.

Below is our point-by-point response.

Reviewer #1

Comment:

“The authors have been adequately considering all the points made in the previous report.”

Response:

We thank the reviewer for the positive assessment and confirmation that the previous comments were satisfactorily addressed. No additional modifications were required.

Reviewer #2

Comment 1 – Formatting inconsistency

“You put an underline in the conclusion section but not in other sections.”

Response:

The formatting inconsistency has been corrected. The heading style is now uniform across all sections.

Comment 2 – Section numbering

“The conclusion and references sections are not numbered.”

Response:

Section numbering has been corrected. The manuscript now includes:

Conclusions

References

Comment 3 – Organization of Materials and Methods

“Why is the materials and methods section not organized? There is not suitable table, etc.”

Response:

The Materials and Methods section has been fully reorganized into clearly numbered subsections (2.1–2.11) for improved clarity and reproducibility.

In addition, Supplementary Table S1 has been added to provide a structured overview of all experimental groups, treatments, doses, timing, and animal allocation. This directly addresses the reviewer’s concern regarding organization.

Comment 4 – Proofreading and minor errors

“The volume of minor errors indicates that the manuscript required more thorough proofreading.”

Response:

The manuscript has undergone comprehensive proofreading. Typographical errors, duplicated numerical expressions, formatting inconsistencies, and minor language issues have been corrected. The revised version has been carefully reviewed to ensure clarity and consistency.

We believe the revised manuscript now fully complies with PLOS ONE formatting and reporting standards. We appreciate the opportunity to improve our work and respectfully submit the revised version for final consideration.

Sincerely,

Dr. Baydaa T. Sih

---

## [Decision Letter · Decision Letter 2]

17 Feb 2026

Differential Modulation of Haematopoietic and Oxidative Injury by PARP-1 and ATR Kinase Inhibition in a Murine Model of Acute Irradiation

PONE-D-25-44594R2

Dear Dr. Sih,

We’re pleased to inform you that your manuscript has been judged scientifically suitable for publication and will be formally accepted for publication once it meets all outstanding technical requirements.

Kind regards,

Hesham M.H. Zakaly, Ph.D.

Academic Editor

PLOS One

Additional Editor Comments (optional):

Reviewers' comments:

Reviewer's Responses to Questions

**Comments to the Author**

Reviewer #2: All comments have been addressed

2. Is the manuscript technically sound, and do the data support the conclusions?

Reviewer #2: Yes

3. Has the statistical analysis been performed appropriately and rigorously?

Reviewer #2: Yes

4. Have the authors made all data underlying the findings in their manuscript fully available?

Reviewer #2: Yes

5. Is the manuscript presented in an intelligible fashion and written in standard English?

Reviewer #2: Yes

Reviewer #2: Dear Autho; the Manuscript is highly improved and deserves to be published in this journal.

Also all corrections are done.

**Do you want your identity to be public for this peer review?** For information about this choice, including consent withdrawal, please see our Privacy Policy

Reviewer #2: No

---

## [Editor Report · Acceptance letter]

PONE-D-25-44594R2

PLOS One

Dear Dr. Sih,

I'm pleased to inform you that your manuscript has been deemed suitable for publication in PLOS One. Congratulations! Your manuscript is now being handed over to our production team.

Kind regards,

on behalf of

Dr. Hesham M.H. Zakaly

Academic Editor

PLOS One